# Formation of memory assemblies through the DNA-sensing TLR9 pathway

Vladimir Jovasevic[1,10], Elizabeth M. Wood[2,10], Ana Cicvaric[2,10], Hui Zhang[2], Zorica Petrovic[2], Anna Carboncino[2], Kendra K. Parker[2], Thomas E. Bassett[2], Maria Moltesen[3,4,5], Naoki Yamawaki[3,4,5], Hande Login[3,4,5], Joanna Kalucka[3,4,5], Farahnaz Sananbenesi[6,7], Xusheng Zhang[8], Andre Fischer[6,7] & Jelena Radulovic[2,3,4,5,9]✉

As hippocampal neurons respond to diverse types of information[1], a subset assembles into microcircuits representing a memory[2]. Those neurons typically undergo energy-intensive molecular adaptations, occasionally resulting in transient DNA damage[3–5]. Here we found discrete clusters of excitatory hippocampal CA1 neurons with persistent double-stranded DNA (dsDNA) breaks, nuclear envelope ruptures and perinuclear release of histone and dsDNA fragments hours after learning. Following these early events, some neurons acquired an inflammatory phenotype involving activation of TLR9 signalling and accumulation of centrosomal DNA damage repair complexes[6]. Neuron-specific knockdown of *Tlr9* impaired memory while blunting contextual fear conditioning-induced changes of gene expression in specific clusters of excitatory CA1 neurons. Notably, TLR9 had an essential role in centrosome function, including DNA damage repair, ciliogenesis and build-up of perineuronal nets. We demonstrate a novel cascade of learning-induced molecular events in discrete neuronal clusters undergoing dsDNA damage and TLR9-mediated repair, resulting in their recruitment to memory circuits. With compromised TLR9 function, this fundamental memory mechanism becomes a gateway to genomic instability and cognitive impairments implicated in accelerated senescence, psychiatric disorders and neurodegenerative disorders. Maintaining the integrity of TLR9 inflammatory signalling thus emerges as a promising preventive strategy for neurocognitive deficits.

Memories of individuals' experiences are represented across assemblies of neurons in hippocampal and cortical circuits. Several mechanisms of formation and maintenance of these assemblies have been proposed. The most prominent such mechanism is stimulus-induced long-term potentiation of synaptic connectivity[7], an energy-demanding process that involves extensive biochemical and morphological adaptations at all levels of neuronal function[8,9]. There is also evidence for contributions of pre-existing developmental and other intrinsic programmes of individual neurons[10], including the baseline expression of the transcriptional factor CREB[11] and lineage of developmental origin[12,13]. Recent focus has also been on the role of the interneuronal perineuronal nets (PNNs) in the stabilization of memory circuits through tightened control of inhibitory inputs to dedicated neuronal assemblies[14]. Here we explored whether an overarching process could integrate stimulus-dependent and pre-existing mechanisms that underlie the commitment of neurons to memory-specific assemblies.

## CFC increases *Tlr9* expression

We first performed an analysis of transcriptional profiles of dorsohippocampal neurons beyond the initial, well-established 24–48 h time window when protein signalling, immediate early gene (IEG) expression and delayed gene expression (for example, growth factors) take place[15,16]. To this end, we performed bulk RNA-sequencing (RNA-seq) of total RNA isolated from individual mouse hippocampi obtained either 96 h or 21 days after contextual fear conditioning (CFC) (one-trial 3 min exposure to a context followed by a 2 s, 0.7 mA shock, constant current), and noted a robust difference in the gene expression profiles for recent relative to remote memory with little within-group variability (a total of 847, with 440 up-regulated) (Fig. 1a). We previously reported that the 21-day gene expression repertoire revolved around cilium and extracellular matrix genes needed for PNN formation[17], but the 96 h gene expression profiles associated with the shaping of recent

[1]Department of Pharmacology, Feinberg School of Medicine, Northwestern University, Chicago, IL, USA. [2]Dominick P. Purpura Department of Neuroscience, Albert Einstein College of Medicine, Bronx, NY, USA. [3]Department of Biomedicine, Aarhus University, Aarhus, Denmark. [4]PROMEMO, Aarhus University, Aarhus, Denmark. [5]DANDRITE, Aarhus University, Aarhus, Denmark. [6]Department for Psychiatry and Psychotherapy, German Center for Neurodegenerative Diseases, University Medical Center, Göttingen, Germany. [7]Cluster of Excellence MBExC, University of Göttingen, Göttingen, Germany. [8]Computational Genomics Core, Albert Einstein College of Medicine, Bronx, NY, USA. [9]Department of Psychiatry and Behavioral Sciences, Psychiatry Research Institute Montefiore Einstein (PRIME), Albert Einstein College of Medicine, Bronx, NY, USA. [10]These authors contributed equally: Vladimir Jovasevic, Elizabeth M. Wood, Ana Cicvaric. ✉e-mail: jelena.radulovic@einsteinmed.edu

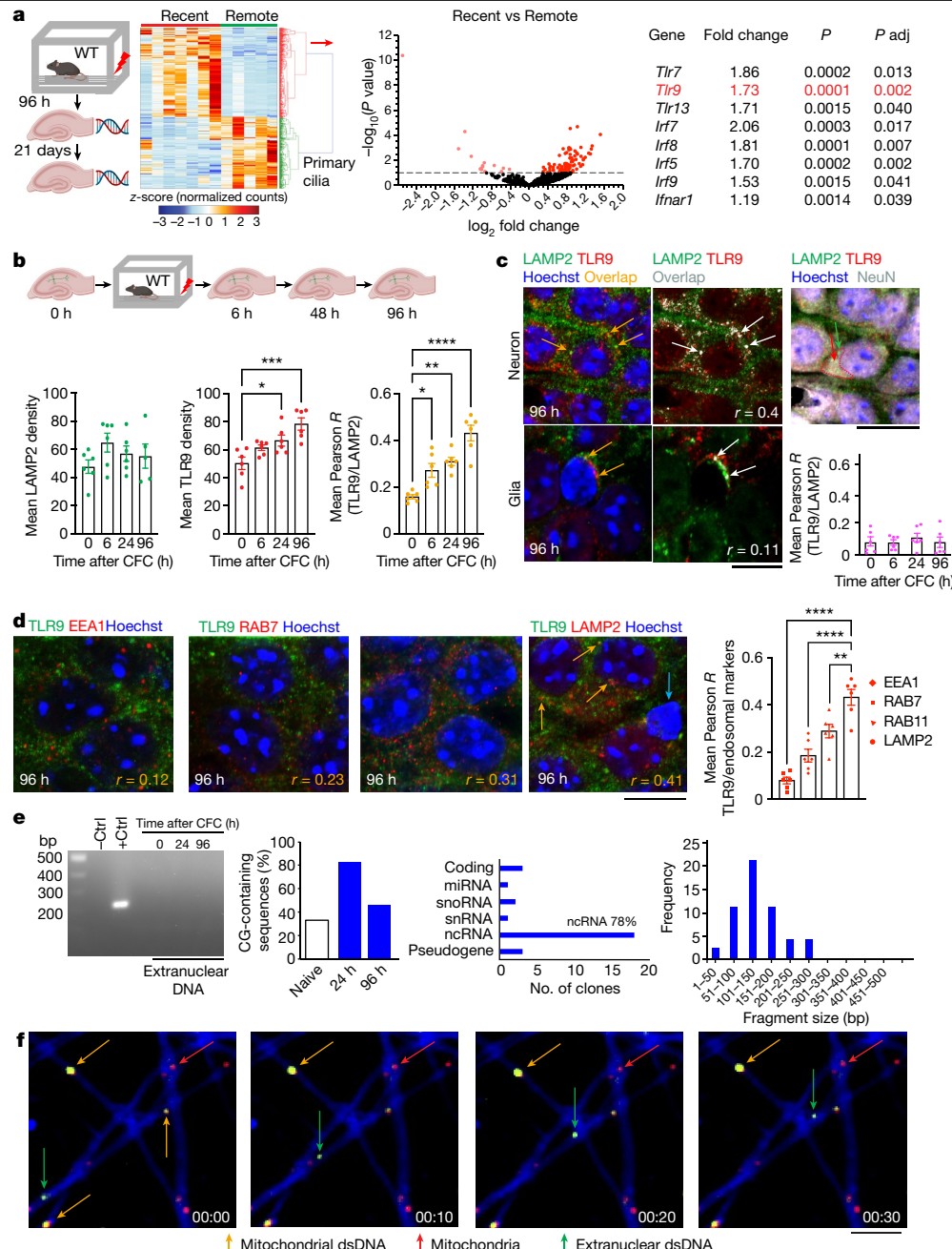

**Fig. 1 | Nucleic acid-sensing activity after CFC. a,** Bulk RNA-seq showed increased expression of 441 genes in hippocampi obtained 96 h after CFC (recent, $n = 7$ mice) compared with those collected 21 days (remote, $n = 5$ mice) after CFC. Volcano plots demonstrate significant increases in expression of genes related to inflammation and TLR signalling. $P$ adj, adjusted $P$ value. **b,** TLR9 protein levels and co-localization of TLR9 with the mature vesicle marker LAMP2 at different times after CFC. LAMP2 levels did not fluctuate, TLR9 levels and its co-localization with LAMP2 increased 6 h after CFC, peaking 96 h later ($n = 6$ mice, 360 neurons per time point; one-way ANOVA; LAMP2: $P = 0.3104$, $F_{(3,19)} = 1.278$; TLR9: $P = 0.0005$, $F_{(3,20)} = 9.363$; co-localization: $P < 0.0001$, $F_{(3,20)} = 21.27$). **c,** TLR9 and LAMP2 signals in glial cells (revealed by nuclear size), show no significant co-localization ($n = 6$ mice, 12 glial cells per time point; one-way ANOVA; $P = 0.8186$, $F_{(3,20)} = 0.3090$). Green arrow, LAMP2; orange and white arrows, LAMP2–TLR9 co-localization; red arrow, TLR9. Scale bars: left, 25 μm; right, 40 μm. **d,** TLR9–vesicle pool co-localization 96 h after CFC (early endosome: EEA1 and RAB7; recycling endosome: RAB11; late endosome: LAMP2) reveals that the highest overlap is with LAMP2 (orange arrows; $n = 6$ mice, 30 neurons per time point; one-way ANOVA; $P < 0.0001$, $F_{(3,20)} = 31.53$). Note the lack of TLR9 and LAMP2 signals in a glial cell (cyan arrow). Data are mean ± s.e.m. Scale bar, 20 μm. **e,** Hippocampal cytosolic dsDNA (naive, 24 h or 96 h after CFC) shows no contamination with nuclear DNA, as revealed by lack of ubiquitous amplification of *Slc17a7* (which encodes vGlut1) (left). Cloning and sequencing identified genomic dsDNA fragments enriched with non-coding gene GC sequences 24 h and 96 h after CFC (left graph), sized 50–300 bp (right graph). Ctrl, control; miRNA, mitochondrial RNA; ncRNA, non-coding RNA; snoRNA, small nucleolar RNA; snRNA, small nuclear RNA. **f,** In vitro imaging of primary hippocampal neurons using fluorescent dyes, revealing mobile extranuclear DNA distinct from mitochondrial DNA (Supplementary Video 1). Scale bar, 10 μm. *$P < 0.05$, **$P < 0.01$, ***$P < 0.001$, ****$P < 0.0001$; NS, not significant; WT, wild type.

memory representations remained unexplored. We found that 96 h after CFC the majority of differentially expressed genes were immune response genes involved in nucleic acid sensing and cytokine release (71

up-regulated and 11 down-regulated) (Fig. 1a and Extended Data Fig. 1). Within immune response genes, the applied analyses identified TLR9 and its downstream NF-κB signalling pathway[18] as the most prominent

functional gene clusters (Extended Data Fig. 1a,b), which is consistent with the reported activation of the NLRP3 inflammasome several days after CFC[19]. We replicated the RNA-seq data with new sets of individual hippocampal samples using quantitative PCR, confirming the robustness of the observed effect (Extended Data Fig. 1c,d). The up-regulation of *Tlr9* gene expression was accompanied by increased TLR9 protein, as well as increased co-localization of TLR9 with LAMP2, a marker of late endosomes and lysosomes (Fig. 1b), an effect that was neuron-specific (Fig. 1c). After CFC, TLR9 co-labelling with LAMP2 was significantly stronger compared with markers of early (EEA1 and RAB7) and recycling (Rab11) endosomes (Fig. 1d), suggesting enhanced TLR9 trafficking to endosomes, enabling DNA recognition and NF-κB activation[20].

TLR9, together with the cyclic GMP–AMP synthase (cGAS)–STING, is the main sensor of extranuclear DNA. In the absence of infection or apoptotic DNA damage, these pathways are typically silent, but under neuronal stress, TLR9 responds to the release of mitochondrial DNA fragments[21]. To determine whether extranuclear dsDNA fragments with TLR9-activating potential can be found in hippocampal neurons, we collected hippocampi of naive mice or mice trained in CFC 24 or 96 h earlier and isolated their extranuclear DNA. After confirming that the fraction was not contaminated with genomic DNA, we proceeded with cloning and sequencing of the isolated dsDNA fragments (Fig. 1e). Contrary to our expectation that, if any, we would identify mitochondrial dsDNA, all cloned dsDNA fragments were of genomic origin, predominantly stemming from non-coding DNA. We cloned a total of 53 dsDNA fragments belonging to 25 unique genomic sequences, none of which corresponded to mitochondrial genes. After CFC, the frequency of CG-containing sequences, which are putative activators of TLR9, transiently increased from 33% to 77%. We also performed live-cell imaging of primary hippocampal neuronal cultures using dyes that detect dsDNA or mitochondrial DNA to distinguish non-mitochondrial from mitochondrial extranuclear DNA. Non-mitochondrial mobile DNA signals were readily detected in the cytosol (Fig. 1f and Supplementary Video 1).

## CFC triggers dsDNA breaks and DDR

Neuronal activity is known to induce transient DNA breaks[3], which are required for the learning-related induction of IEGs. However, such breaks occur in enhancers of coding genes and are repaired within minutes without affecting neuronal homeostasis[5]. To examine the origin of extranuclear dsDNA fragments generated by CFC, we hypothesized that, in discrete neuronal populations, activity-induced DNA damage might be more substantial and sustained. We performed a time-course study up to 96 h after CFC, focusing on dsDNA breaks in individual cells of the hippocampal CA1 region because of its well-established role in the formation and retention of context memories[22]. To detect DNA damage, we performed immunofluorescent labelling with antibodies specific for the dsDNA break-binding phospho-histone γH2AX[3] and quantified the signals using plot profile or particle analyses. To minimize the interference of background γH2AX activity unrelated to dsDNA breaks, foci were selected if their fluorescence intensity was at least two standard deviations greater than the average intensity of the total nuclear signal for each CA1 region of interest. One and three hours after CFC, we identified discrete, patchy clusters of CA1 nuclei exhibiting a significant increase of the total number of neurons with clear γH2AX foci (Fig. 2a, left and Supplementary Video 2). Whereas the number of nuclear foci subsequently decreased, much larger γH2AX-labelled foci emerged in individual neurons, persisting from 6 to 96 h (Fig. 2a, middle). All γH2AX signals were neuron-specific and were not found in astrocytes or microglia (Fig. 2a, right). Co-labelling with the nuclear envelope protein lamin B1 revealed that at the time of maximal dsDNA break detection, a subset of nuclei exhibited envelope ruptures (discontinuation of lamin B1 labelling), resulting in perinuclear release of γH2AX in RNA-rich areas typical of the endoplasmic reticulum

(Fig. 2b and Extended Data Fig. 2a), a primary localization site of inactive TLR9. The number of ruptures increased significantly 1 h after CFC and remained detectable in a smaller number of nuclei throughout the 96 h period. The perinuclear γH2AX signals co-labelled with TLR9 (Fig. 2c) and to a lesser extent with the DNA dye Hoechst and with antibodies recognizing dsDNA (Extended Data Fig. 2b). These findings indicated that in some neurons, γH2AX and dsDNA, alone or in complexes, were released from the nucleus in TLR9-containing perinuclear sites. The persistent γH2AX signals found at later time points (6–96 h) had a diameter of 4 μm or larger, and co-localized with the centrosome markers centrin and γ-tubulin, demonstrating pericentrosomal localization rather than localization at sites of dsDNA breaks (Fig. 2d, left), as suggested previously[4]. Contrary to the small, sharp and focused nuclear signals, all pericentrosomal signals were large and fuzzy. These signals were strongly and consistently co-labelled with 53BP1, the main mediator of DNA repair by nonhomologous end joining[23,24] (Fig. 2d, right). Neurons undergoing dsDNA breaks and DNA damage repair (DDR) did not show any nuclear morphological indices of apoptosis. Both nuclear and pericentrosomal γH2AX signals overlapped with cleaved caspase-3 foci, but the patterns of their localization were consistent with the non-apoptotic, probably memory-related roles of cleaved caspase-3[25] (data not shown). Although consistent with evidence of centrosomal localization of 53BP1[26] and with increasing recognition of centrosomal control of DDR[27], the extent to which dsDNA breaks and centrosomes shared molecular components, including γH2AX, dsDNA fragments and DNA repair enzymes was unexpected and suggested that the proposed role of centrosomes in the maintenance of gene integrity in dividing cells[28] also applies to adult neurons undergoing memory-related activity.

We next studied whether the detection of γH2AX signals was related to the CFC-induced up-regulation of IEGs. We collected brain sections 1 h after CFC because this time point is optimal for the detection of Fos in CA1[29], as well as EGR1, which shows both sizeable baseline expression as well as up-regulation after learning[30]. We also performed immunostaining for CREB, whose baseline expression has been directly implicated in memory[11]. Many neurons with the typical, homogenous nuclear labelling for Fos, CREB and EGR1 were devoid of γH2AX signals and showed overall insignificant correlation (Fig. 2e and Extended Data Fig. 3). Even in instances of co-detection, IEG labelling in neurons positive for γH2AX was mainly punctate and scattered. Only 20% of all γH2AX-positive nuclei were Fos-positive (52% of Fos+ nuclei were co-labelled with γH2AX). Instead, γH2AX signals were associated with inflammatory signalling, as shown by a high degree of co-localization with RELA, the most abundant NF-κB family member, which translocates to the nucleus upon activation (Extended Data Fig. 3).

We also examined to what extent IEG or DDR (centrosomal γH2AX signals)-responsive neurons up-regulate Fos during subsequent memory reactivation (Fig. 2f and Extended Data Fig. 3e,f). Because IEG responses peak 1 h after CFC and DDR peaks at 96 h, we permanently labelled the CFC-activated IEG population with GFP using the robust activity marker PRAM–GFP driven by the Fos promoter[31]. In this way we could examine Fos reactivation 96 h after CFC in both populations by comparing the overlap of γH2AX, GFP and Fos. As γH2AX+Fos+ neurons were scarce, we compared 30–34 neurons per population. Relative to PRAM-labelled neurons, γH2AX+ (DDR population) neurons showed significantly smaller Fos responses after memory reactivation, demonstrating that both after CFC and after memory reactivation, IEG and inflammatory signalling occurred predominantly in non-overlapping neuronal populations.

## *Tlr9* in CA1 neurons is required for context memory

We next investigated whether the inflammatory response is a side effect of learning-induced DNA damage or whether it contributes to memory formation. We induced a neuron-specific knockout of TLR9

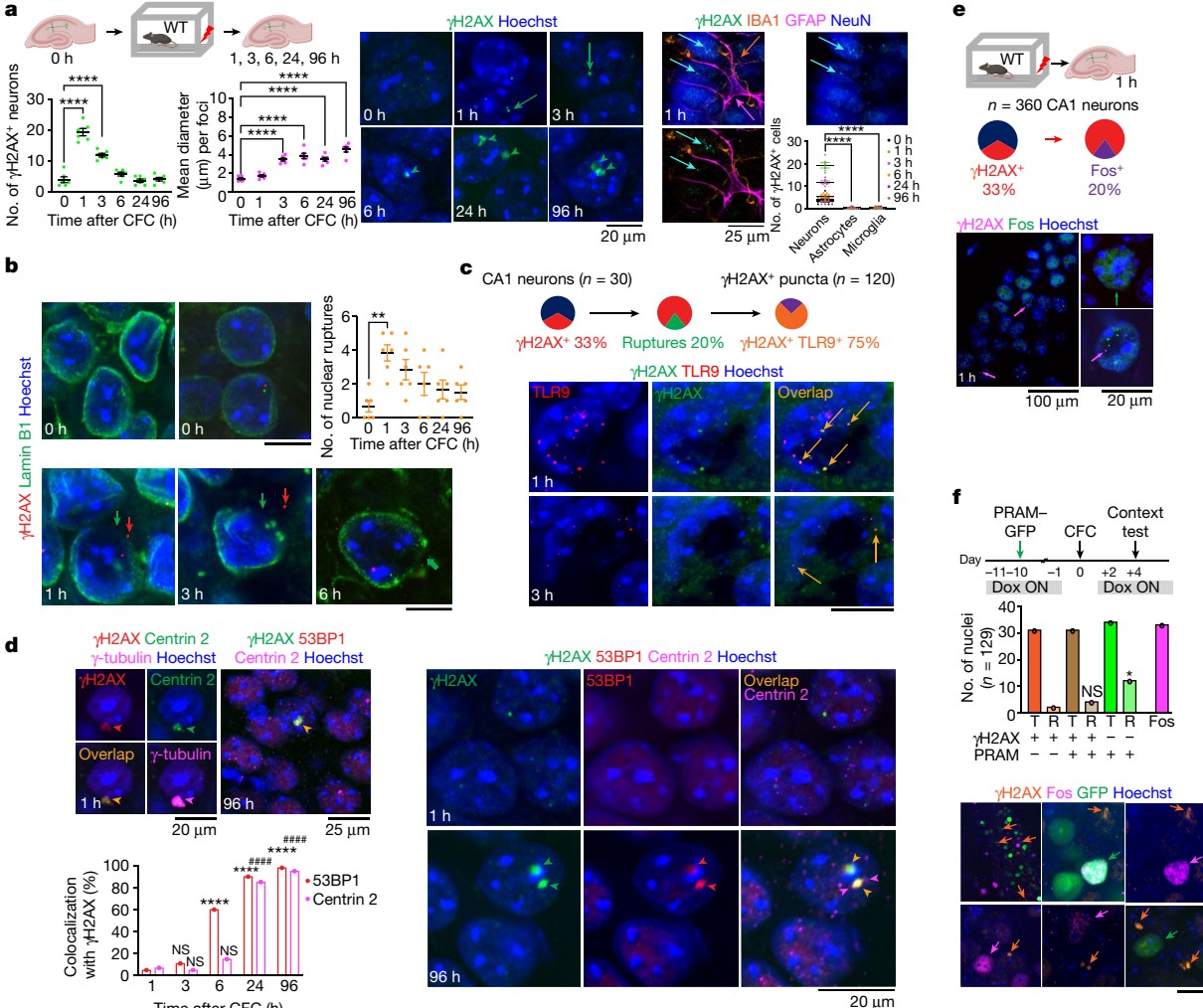

**Fig. 2 | DNA damage and DDR after CFC. a**, Left, number of neurons showing γH2AX puncta ($n = 360$ neurons per group; one-way ANOVA, $P < 0.0001$, $F_{(5,30)} = 65.09$) and size of γH2AX foci ($n = 360$ neurons per group; one-way ANOVA; $P < 0.0001$, $F_{(5,30)} = 38.17$) after CFC. Right, localization of γH2AX (green arrows) in neurons (marked by NeuN; cyan arrows) relative to astrocytes (marked by GFAP; purple arrows) and microglia (marked by IBA1; orange arrows) (two-way ANOVA; factor: cell type, $P < 0.0001$, $F_{(2,90)} = 673.8$; factor: time, $P < 0.0001$, $F_{(5,90)} = 77.73$; cell type × time, $P < 0.0001$, $F_{(10,90)} = 77.95$). Bottom right, number of γH2AX foci. **b**, Nuclear envelope ruptures coinciding with detection of extranuclear γH2AX (red arrows) and DNA (green arrows) ($n = 6$ mice; one-way ANOVA; $P = 0.0038$, $F_{(5,30)} = 4.445$). Scale bars: top row, 25 μm; bottom row, 20 μm. **c**, Extranuclear γH2AX overlapping with TLR9 (orange arrows) ($n = 120$ total, 75% overlap). Scale bar, 20 μm. **d**, Pericentrosomal accumulation of γH2AX shown by co-localization with centrin 2 and γ-tubulin (arrowheads). Additional co-recruitment of 53BP1, revealing centrosomal DDR ($n = 30–131$ neurons; two-tailed Chi-square test; $\chi^2_{(4)} = 22.98$, $P < 0.0001$; post hoc analysis using Bonferroni-corrected $\alpha = 0.05$, 53BP1: 3 h versus 1 h $^{NS}P = 0.116$, 6 h versus 1 h $^{****}P < 0.0001$, 24 h versus 1 h $^{****}P < 0.0001$, 96 h versus 1 h $^{****}P < 0.0001$; centrin 2: 3 h versus 1 h $^{NS}P = 0.5061$, 6 h versus 1 h $^{NS}P = 0.2061$, 24 h versus 1 h $^{####}P < 0.0001$, 96 h versus 1 h $^{####}P < 0.0001$; adjusted $\alpha P < 0.001$). **e**, Co-labelling of γH2AX+ (purple arrows) and Fos+ (green arrow) neurons (20%). **f**, Significantly lower number of γH2AX+ neurons (orange arrows) relative to PRAM+ neurons (green arrows) show memory reactivation (co-labelling with Fos; purple arrows) ($n = 216$ neurons; two-tailed Chi-square test; $\chi^2_{(3)} = 6.518$, $P = 0.0384$; post hoc analysis using Bonferroni-corrected $\alpha = 0.05$, γH2AX+ versus PRAM+ $^*P = 0.0215$, γH2AX+PRAM+ versus PRAM+ $^{NS}P = 0.1007$; adjusted $\alpha P < 0.025$). Data are mean ± s.e.m. Dox, doxycycline; T, total neurons; R, reactivated neurons. Scale bar, 20 μm.

in CA1 dorso-hippocampal neurons of *Tlr9*$^{fl/fl}$ mice by locally injecting adeno-associated virus (AAV9) expressing Cre recombinase–GFP or GFP under control of the human synapsin promoter (*Syn*) (Fig. 3a, left). The *Syn-cre*-injected mice showed impaired context memory, as revealed by significant reduction of freezing behaviour over multiple tests (Fig. 3a, middle). The neuron-specific virus expression was shown by differential staining for astrocytes and microglia (Fig. 3a, right and Extended Data Fig. 4a). Knockdown was validated by quantitative analysis of TLR9 and RELA (Fig. 3b). The memory deficit was replicated by a different approach, using injection of AAV9 expressing *Syn*-driven *Tlr9*-targeting short hairpin RNA (shRNA) in the hippocampi of wild-type mice (Fig. 3c). The hippocampal *Tlr9* knockdown (*Tlr9*-KO) also impaired trace fear conditioning (TFC), as shown by reduced freezing during context and

tone tests, but did not affect delay fear conditioning (DFC), a form of hippocampus-independent learning (Fig. 3d). Because viral injections are known to cause inflammatory responses of astrocytes and microglia (as confirmed in our post-mortem analysis; Extended Data Fig. 4b), we also determined the contribution of these cell populations to the observed memory deficits. Astrocytic knockdown using *GFAP-cre* did not affect CFC after injection in *Tlr9*$^{fl/fl}$ mice (Extended Data Fig. 4c). Consistent with previous findings[32], microglial depletion was similarly ineffective, as determined in both wild-type and *Tlr9*$^{fl/fl}$ mice receiving a CSF1R inhibitor in their diet or regular diet after injection of *Syn-cre* (Extended Data Fig. 4d,e). Introduction of *Syn-cre* in the hippocampi of WT mice did not affect CFC relative to *Syn-GFP*-injected wild-type controls (Extended Data Fig. 4e, bottom right).

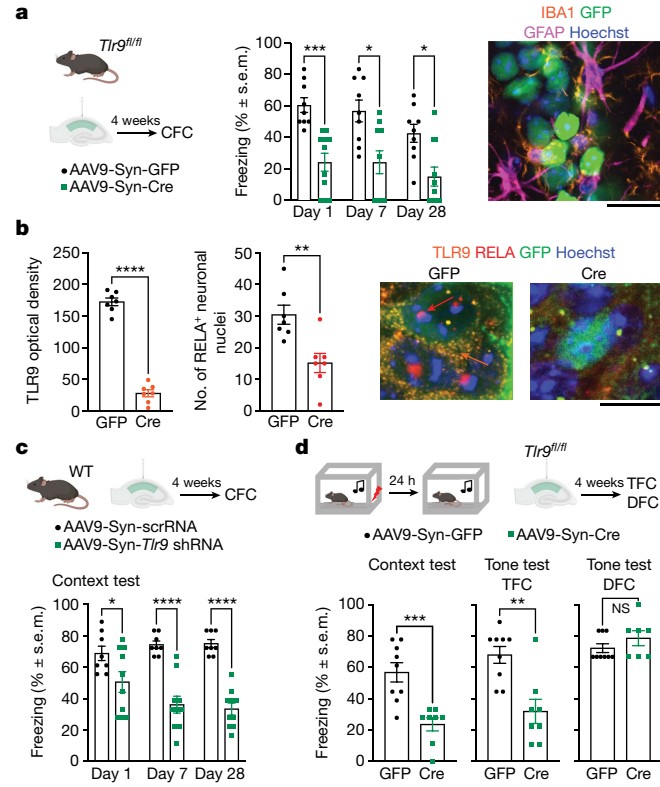

**Fig. 3 | Impaired context memory after neuron-specific deletion of hippocampal _Tlr9_. a**, Left, experimental schematic. Middle, persistent reduction of freezing during context tests of _Tlr9_$^{fl/fl}$ mice injected intrahippocampally with _Syn-cre_ ($n = 11$ mice) compared with the control group injected with _Syn-GFP_ ($n = 9$ mice; two-way ANOVA with repeated measures; factor: virus, $P = 0.0007$, $F_{(1,18)} = 16.54$; factor: test, $P = 0.0007$, $F_{(1.936,34.84)} = 9.30$, virus × test, $P = 0.4358$, $F_{(2,36)} = 0.85$). Right, lack of co-localization of _Syn-cre_ with astrocytic and microglial markers. Scale bar, 40 µm. **b**, Left, reduction of TLR9 levels (mean optical density per neuron, 60 neurons per mouse, 7 mice per group; two-tailed unpaired _t_-test; $t_{12} = 17.4700$, $P < 0.0001$) and RELA nuclear signal (60 neurons per mouse, 7 mice per group; two-tailed unpaired _t_-test; $t_{12} = 3.5679$, $P = 0.0039$) after neuron-specific deletion of TLR9. Right, representative micrographs. Orange arrow indicates TLR9, red arrow indicates RELA. Scale bar, 20 µm. **c**, Persistent reduction of freezing during context tests of wild-type mice injected intrahippocampally with neuron-specific _Tlr9_ shRNA ($n = 10$ mice) compared with scrambled RNA (scrRNA) ($n = 8$ mice). Two-way ANOVA with repeated measures; factor: virus, $P < 0.0001$, $F_{(1,16)} = 35.50$; factor: test, $P = 0.2347$, $F_{(2,32)} = 1.517$; virus × test, $P = 0.0027$, $F_{(2,32)} = 7.168$. **d**, After TFC, _Tlr9_-KO resulted in impaired freezing during context (two-tailed unpaired _t_-test; $t_{15} = 4.362$, $P = 0.0006$) and tone tests after TFC (two-tailed unpaired _t_-test; $t_{15} = 3.899$, $P = 0.0014$) (GFP, $n = 9$ mice; _Syn-cre_, $n = 7$ mice), but intact freezing during the tone test after DFC (two-tailed unpaired _t_-test; $t_{14} = 1.214$, $P = 0.2448$). Data are mean ± s.e.m.

We next examined the involvement of TLR9 relative to cGAS–STING signalling in memory using pharmacological manipulations of these pathways. The TLR9 antagonist oligonucleotide ODN2088 significantly impaired CFC, whereas the small drug cGAS–STING inhibitors RU-521 and H-151 were ineffective relative to vehicle (Extended Data Fig. 5a). Consistent with these observations, CFC was intact in mice lacking the _Sting1_ gene relative to wild-type controls (Extended Data Fig. 5b). Finally, to test the role of endogenous processing of extranuclear dsDNA in memory formation, we also manipulated the levels of two DNases controlling cellular DNA sensing. DNase2 digests dsDNA in shorter dsDNA fragments required for TLR9 binding and activation[33]. To prevent the generation of TLR9-activating DNA fragments we virally overexpressed _Dnase2_ shRNA or scrambled shRNA in mouse dorsal

hippocampi. _Dnase2_ shRNA treatment significantly impaired memory formation (Extended Data Fig. 5c). TREX1 is a cytosolic DNase that predominantly restricts the activity of the cGAS–STING pathway[33]. We disrupted cGAS–STING-mediated DNA sensing by overexpressing TREX1–GFP in the dorsal hippocampus, using GFP overexpression as a control. In these mice, context memory formation was not affected (Extended Data Fig. 5d). In sum, these experiments provided converging evidence for a role of neuron-specific TLR9-mediated but not cGAS–STING-mediated dsDNA sensing in the formation and persistence of context memories.

## _Tlr9_ knockdown disrupts CFC-induced gene expression

To better characterize the effects of CFC and neuron-specific _Tlr9_-KO on gene expression in individual neuronal and non-neuronal hippocampal populations, we performed single-nucleus RNA-sequencing (snRNA-seq). For this experiment, we expressed the viral vectors in the entire dorsal hippocampus to analyse their effects across all subfields. We collected nuclei and prepared libraries (Lib) from 4 groups of mice (pooled from 5 mice per group): _Tlr9_$^{fl/fl}$ mice injected with _Syn-GFP_ and euthanized 96 h after CFC (Lib 1); _Tlr9_$^{fl/fl}$ mice injected with _Syn-cre_ and euthanized 96 h after CFC (Lib 2); naive _Tlr9_$^{fl/fl}$ mice injected with _Syn-GFP_ (Lib 3); and naive _Tlr9_$^{fl/fl}$ mice injected with _Syn-cre_ (Lib 4). Lib 4 was sorted to obtain GFP-positive and GFP-negative nuclei and used to determine the cell specificity of _cre_ expression by snRNA-seq (Extended Data Fig. 6a), whereas the other samples were analysed for CFC-induced (Lib 1 versus Lib 3) and _Tlr9_-KO (Lib 2 versus Lib 1) effects (Fig. 4). An unsupervised algorithm identified 29 clusters, with the highest diversity seen among excitatory CA1 neurons (12 clusters) relative to dentate gyrus granule cells (DGGC) (4 clusters), interneurons (4 clusters) and non-neuronal cells (typically 1 cluster each) (Extended Data Fig. 6b). Analyses of robust gene expression changes (more than 1.5-fold) revealed both increases and decreases of gene expression in the CFC and _Tlr9_-KO groups (Fig. 4a and Supplementary Tables 1 and 2). _Tlr9_-KO abolished the CFC-induced gene expression but did not affect the expression of these genes relative to the naive control (Lib 2 versus Lib 3; Supplementary Table 3). An exception was cluster 26, which showed a paradoxical up-regulation of genes involved in axon guidance and adhesion in _Tlr9_-KO (Extended Data Fig. 7). CFC induced a set of highly conserved genes across neuronal clusters and occasionally in non-neuronal cells, with most genes associated with the endomembrane system and acting as endoplasmic reticulum chaperones and regulators of vesicle trafficking and function, and with interleukin-6 production (Fig. 4b and Supplementary Tables 4 and 5). Some of the proteins encoded by these genes mediate TLR9 folding (HSP90B1) and induce vesicle acidification (ATP6V0C), which are critical for TLR9 activation (Fig. 4c and Extended Data Fig. 6c), whereas others, such as those associated with interleukin-6 signalling, require TLR9 activation. In line with recent reports, we detected many doublecortin-positive (DCX$^+$) neurons among CA1 neurons, DGGC and interneurons[34] (Fig. 4d and Extended Data Fig. 8), and found that the conserved gene responses were more pronounced in the DCX$^+$ cluster (Fig. 4e). The expression of _Dcx_ in CA1 and up-regulation of _Hsp90b1_ after CFC were also confirmed by RNAscope analysis (Extended Data Fig. 9). A reactome analysis of broader gene expression profiles (including all genes showing a significant increase in expression ($P < 0.05$) in addition to those showing conserved changes) revealed CFC-induced gene expression across pathways involved in RNA and protein metabolism, vesicle trafficking, immunity, cell cycle, DNA repair and cilium assembly, however in this case, DCX$^-$ excitatory CA1 and DGGC neurons were the most affected population (Fig. 4f). Although we did not detect low abundance transcripts of immune mediators (such as TLR9 or RELA), the findings confirmed that 96 h after CFC, neurons acquire inflammatory phenotypes associated with TLR signalling, DNA repair, ciliogenesis and vesicle trafficking. In addition to the described effects,

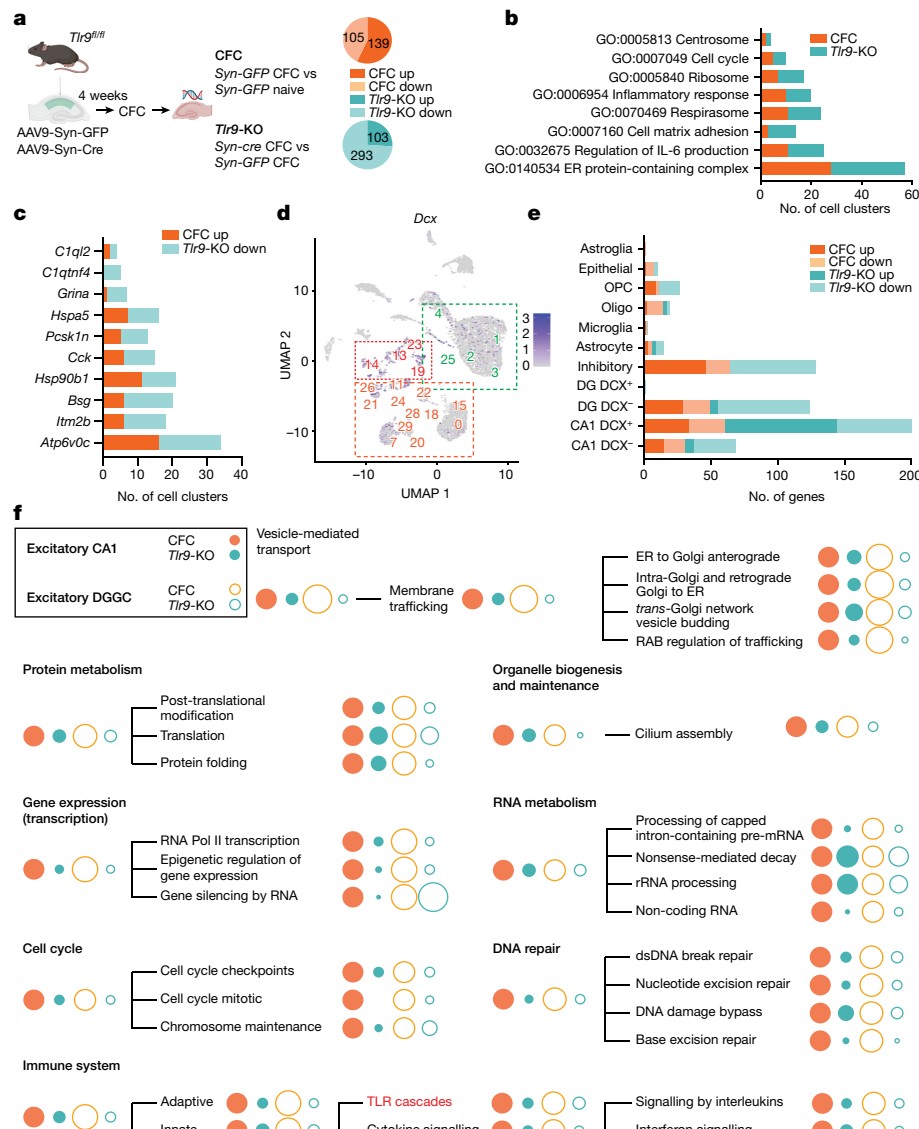

**Fig. 4 | Single-cell changes of gene expression after neuron-specific deletion of hippocampal *Tlr9*. a**, *Tlr9^fl/fl^* mice injected with *Syn-cre* or *Syn-GFP* were trained in CFC or left untrained (naive), and 96 h later, dorsal hippocampal nuclei were isolated and processed for snRNA-seq. Robust (more than 1.5-fold) changes of gene expression were found after CFC in mice injected with control virus (*Syn-GFP* CFC versus *Syn-GFP* naive) or Cre virus (*Syn-cre* CFC versus *Syn-GFP* CFC). **b**, Gene Ontology (GO) analysis reveals that most genes regulated by CFC and *Tlr9*-KO involve endoplasmic reticulum (ER), mitochondrial function, IL-6 production and inflammation. CFC induced up-regulation of these genes, whereas *Tlr9*-KO blocked this effect. **c**, The most conserved genes up-regulated by CFC and down-regulated by *Tlr9*-KO across cell clusters include *Atp6v0c* and *Hsp90b1*, key regulators of TLR9 function (Extended Data Fig. 6). **d**, Dcx expression superimposed on uniform manifold approximation and projection (UMAP) analysis of snRNA-seq data from dorsal hippocampal cells. The expression of *Dcx* in the main neuronal clusters is outlined in orange (excitatory CA1), red (inhibitory) and green (DGGC). **e**, Cell-specific changes in gene expression demonstrates dominant effects of CFC and *Tlr9*-KO on gene expression in neurons relative to other cell populations with particularly strong effects of *Tlr9*-KO in DCX^+ CA1 neurons. DG, dentate gyrus; oligo, oligodendrocyte; OPC, oligodendrocyte precursor cell. **f**, Reactome analysis reveals the major functional gene networks affected by CFC and *Tlr9*-KO in DCX^– CA1 and DGGC neurons. Circles are scaled to the percentage effect of *Tlr9*-KO on the gene expression or pathway relative to *Syn-GFP* with CFC. TLR cascades, DDR and cilium assembly are enriched among the pathways that are most up-regulated by CFC and down-regulated by *Tlr9*-KO. RNA Pol II, RNA polymerase II.

CFC and *Tlr9*-KO induced some unexpected phenotypic changes of excitatory neurons, such as fluctuations of vGlut2 and DCX (Extended Data Fig. 8).

Given that immune cells express high levels of TLR9 and are also a source of circulating, cell-free DNA[35], we also examined whether blood-borne infiltrating cells and DNA from extracellular sources might contribute to the observed up-regulation of TLR9 signalling and memory. Such contribution was unlikely, however, given the absence of lymphocytic and myeloid cell markers in our samples (Extended Data Fig. 10a). Similarly, systemic or intrahippocampal infusions of DNase1 (which efficiently degrades cell-free DNA) before and shortly after CFC were ineffective (Extended Data Fig. 10b), as was intrahippocampal injection of DNase1 and S1 nuclease, which degrades single-stranded DNA, before the context test.

In summary, the snRNA-seq approach identified ongoing CFC-induced cell-specific (mainly neuron-specific) gene expression responses and discrete phenotypic changes 96 h after CFC. *Tlr9*-KO blunted the induction of most of the up-regulated genes without affecting CFC down-regulated genes and induced additional, CFC-unrelated changes, especially in excitatory cluster 26.

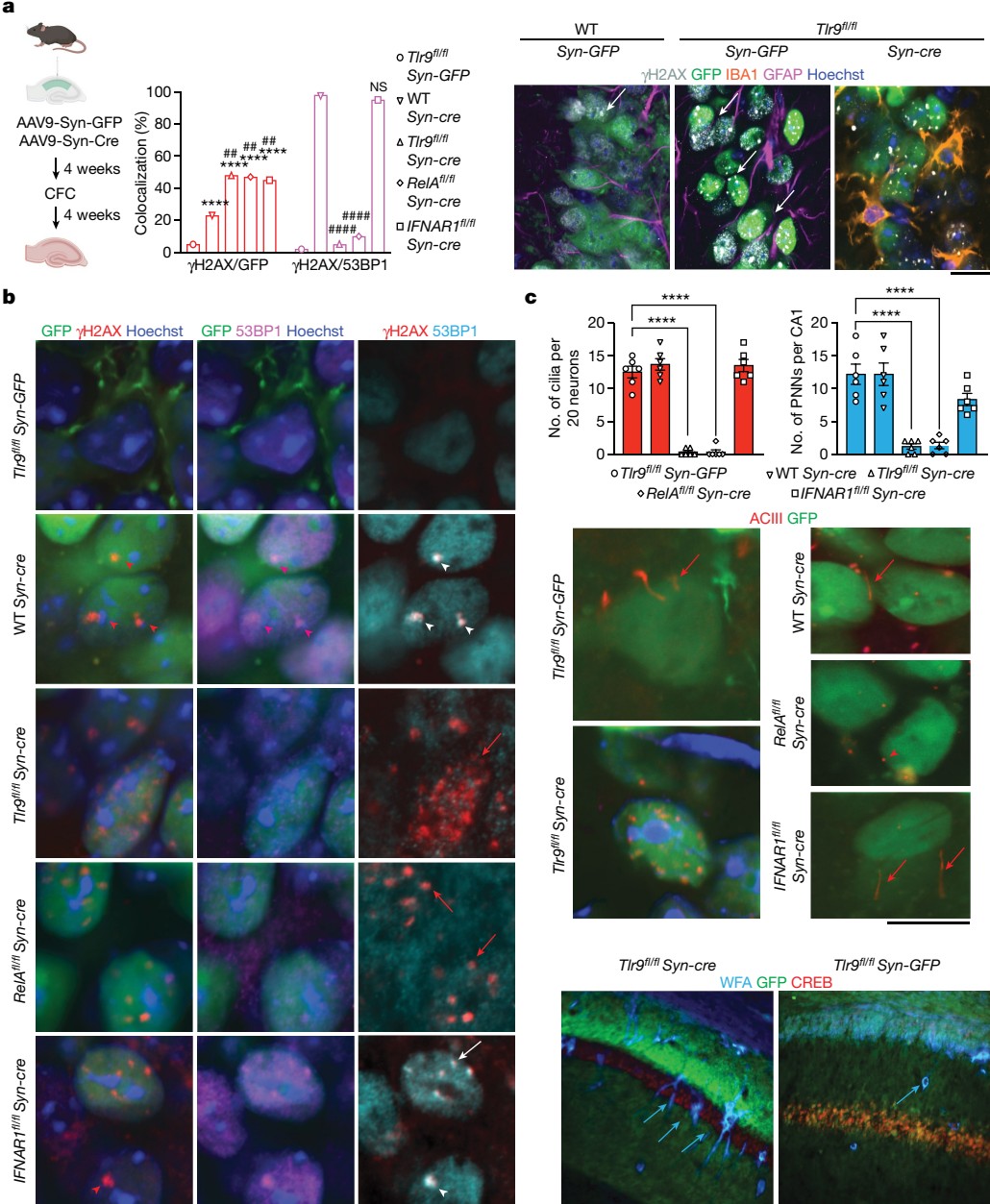

**Fig. 5 | Impaired DDR, ciliogenesis and PNN formation by neuron-specific deletion of hippocampal *Tlr9*. a**, Increased number of neurons showing γH2AX signals in mice hippocampally injected with *Syn-cre* relative to *Syn-GFP* (*n* = 5 mice (150 neurons) per group; two-tailed Chi-square test; $\chi^2_{(4)} = 54.86$, *P* < 0.0001; post hoc analysis using Bonferroni correction, *α* = 0.05; versus GFP: WT *cre* \*\*\*\**P* < 0.0001, *Tlr9 fl/fl* \*\*\*\**P* < 0.0001, *Rela fl/fl* \*\*\*\**P* < 0.0001 and *Ifnar1 fl/fl* \*\*\*\**P* < 0.0001). This effect was further potentiated by injection of *Syn-cre* in hippocampi of *Tlr9 fl/fl*, *Rela fl/fl* and *Ifnar1 fl/fl* mice (versus WT *cre*: *Tlr9 fl/fl* ##*P* < 0.002, *Rela fl/fl* ##*P* < 0.003 and *Ifnar1 fl/fl* ##*P* < 0.006; adjusted *α P* < 0.001). The observed genomic instability was accompanied by centrosomal DDR in wild-type and *Ifnar1 fl/fl* mice, but blunted in *Tlr9 fl/fl* and *Rela fl/fl* mice injected with *Syn-cre* (*n* = 5 mice (150 neurons) per group; two-tailed Chi-square test; $\chi^2_{(4)} = 124.1$, *P* < 0.0001; post hoc analysis using Bonferroni correction *α* = 0.05;

versus WT *cre*: *Tlr9 fl/fl* ####*P* < 0.0001, *Rela fl/fl* ####*P* < 0.0001 and *Ifnar1 fl/fl* NS*P* = 0.9681). Right, up-regulated γH2AX signals (white arrows) were seen in CA1 neurons but not in adjacent astrocytes or microglia. Scale bar, 20 μm. **b**, Illustration of the findings presented in **a**. To facilitate signal detection of γH2AX–53BP1 overlap, 53BP1 images were pseudocoloured with cyan. Scale bar, 20 μm. **c**, Top, whereas *Syn-cre* injection in wild-type mice and *Ifnar1* deletion did not affect the number of cilia (*n* = 6 mice per group, 30 neurons per mouse) and PNNs (*n* = 6 mice per group, one slice of dorsal CA1 per mouse), *Tlr9* and *Rela* knockout impaired ciliogenesis and PNN formation (top; one-way ANOVA; ACIII: *P* < 0.0001, $F_{(4,25)} = 97.47$; PNNs: *P* < 0.0001, $F_{(4,25)} = 23.31$). Representative micrographs depicting ACIII signals (middle, red arrows; scale bar, 20 μm) and PNNs (bottom, blue arrows; scale bar, 100 μm). *Wisteria floribunda* lectin (WFN). Data are mean ± s.e.m.

## TLR9 controls DDR, ciliogenesis and PNN build-up

To determine the cellular consequences of *Tlr9*-KO, we collected the hippocampi of WT and *Tlr9 fl/fl* mice 24 h after their last memory test following each experiment, and first examined DNA damage and DDR response. In addition, we collected hippocampi of *Rela fl/fl* and interferon

receptor 1-floxed (*Ifnar1 fl/fl*) mice undergoing the same genetic and behavioural manipulations to examine the potential contributions of these pathways downstream of TLR9. A control group that was not exposed to CFC was also included. We found a significant increase of the number of neurons with dsDNA breaks in all mice, including wild-type mice, injected with *Syn-cre*. However, this effect was more

pronounced in the genetically modified lines (Fig. 5a and Extended Data Fig. 11) and was also found in mice that were not exposed to CFC. There were large multifocal nuclear accumulations of γH2AX and 53BP1 in wild-type and *Ifnar1*[fl/fl] mice injected with *Syn-cre*, as previously found in nuclear bodies formed around DNA lesions in cell lines undergoing replication stress[28], with an average size of 2 μm corresponding to nuclear stress bodies[36]. Notably, in *Tlr9*[fl/fl] and *Rela*[fl/fl] mice, *Syn-cre* completely disrupted the recruitment of 53BP1 to sites of DNA damage and to centrosomal DDR sites (Fig. 5a,b). To examine whether this affected other memory-related centrosomal functions, we also examined the consequences of individual gene knockdown on ciliogenesis and cilium-dependent PNN formation[17]. Whereas wild-type and *IFNAR1*[fl/fl] mice injected with Syn-Cre showed intact ciliogenesis in CA1 neurons, as shown by filamentous staining for adenyl cyclase III (ACIII), neurons of *Tlr9*[fl/fl] and *Rela*[fl/fl] mice showed punctate and disorganized ACIII labelling accompanied by disappearance of PNNs, with the scarce remaining PNNs showing markedly reduced complexity (Fig. 5c and Extended Data Fig. 12). These findings demonstrated that lack of TLR9 disrupts the nuclear and centrosomal DDR machinery so that at the time of CFC, CA1 neurons could not recruit DDR complexes or form cilia and PNNs. Whereas both *Rela*-KO and *Ifnar1*-KO could contribute to *Tlr9*-KO-induced genomic instability, the effects on 53BP1, ciliogenesis and PNN formation were most likely to be mediated by the RELA downstream pathway.

## Discussion

The recruitment of individual neurons to assemblies is essential not only for encoding individual memories, but also for protecting them from streams of incoming information over time, ensuring stability and persistence of memory representations. On the basis of the evidence presented here, we suggest that in distinct populations of hippocampal CA1 excitatory neurons, this is achieved through learning-induced TLR9 signalling linking DNA damage to DDR. Over several days, such neurons acquired an inflammatory phenotype involving activation of the TLR9 DNA-sensing pathway and pericentrosomal accumulation of DDR complexes. TLR9 activation was most probably triggered by the release of γH2AX and dsDNA fragments, stemming predominantly from non-coding DNA, into the endomembrane system. Some of these neurons underwent more profound phenotypic fluctuations consistent with changes of their differentiation state. The identified genetic, molecular and cellular phenotypes were blocked by neuron-specific knockdown of *Tlr9*, and some, including disruption of pericentrosomal DDR and ciliogenesis, were replicated by *Rela* knockdown but not by *Ifnar1* knockdown, identifying NF-κB rather than interferon pathways as downstream mediators. The involvement of DNA sensing by TLR9, but not the evolutionary older cGAS–STING pathway in CFC, suggests that neurons have adopted an immune-based memory mechanism involving trafficking of dsDNA and histone to endolysosomes, rather than their release in the cytosol[37]. This is consistent with the gene expression profiles observed 96 h after CFC, which were primarily related to endoplasmic reticulum function and vesicle trafficking.

Although we could not directly demonstrate activation of TLR9 by specific DNA fragments at the single-neuron level, we found evidence that several TLR9-activating mechanisms are induced by CFC. TLR9 signalling is induced mainly by unmethylated CpG DNA sequences, which are predominantly of bacterial origin[38], but such sequences can also be generated through demethylation of mammalian DNA during learning and long-term potentiation[15]. Moreover, there is increasing evidence of TLR9 activation by mammalian self-DNA[39,40] and by histones[41], both of which we found to be released from the nucleus following nuclear envelope ruptures.

Given the association of dsDNA damage with neurodegeneration[42], neurons undergoing learning-induced dsDNA breaks might be expected to be excluded from memory assemblies. This was also suggested by the lack of significant association between the observed inflammatory responses with other mechanisms implicated in learning, including IEG responses. However, we found that these neurons were required for memory formation, but only if their TLR9-mediated DDR machinery was intact. Although many factors could predispose individual neurons to respond with dsDNA damage rather than IEG, it needs to be considered that diverse responses to synaptic input[43] could be based on the genetic diversity of neurons generated throughout life[44]. The occurrence of sustained dsDNA breaks might reflect a particularly intense CA1 response of discrete CA1 neurons best tuned to the context information carried by excitatory input[5], and resulting in their lasting representation of that context. A shared history of cycles of DNA damage and repair could similarly organize neuronal assemblies by collective experience as a shared history of IEG response or birthdate, providing yet another dimension to memory organization. An advantage of recruiting distinct neuronal populations to memory representations is that they can uniquely contribute to their stability, specificity, flexibility and other properties[31,45]. Whereas populations displaying IEG responses during memory formation and reactivation might be better suited for retrieval-mediated memory updates and modifications[46], the population displaying an inflammatory DDR phenotype seems better suited to maintain stable representations of context memories. By coordinating centrosome-mediated DDR, TLR9 could integrate genomic responses to synaptic inputs (dsDNA breaks), which subserve memory specificity, with delayed, extrasynaptic memory mechanisms (ciliogenesis and PNN formation), which promote memory persistence[14,17,40], at least when memories are hippocampally dependent[47].

Our findings support recent data revealing more widespread expression of DCX in hippocampal and cortical neurons than originally believed[34,48]. Fluctuations of DCX in response to hyperexcitation and inflammation are thought to reflect changes in neuronal maturation states[49]. We found similar changes within small, DCX+ CA1 clusters (clusters 11, 21 and 24) that lost their DCX+ phenotype in response to CFC, indicating that some neurons, at least transiently, shifted to a more mature state. Given the essential role of DCX in nucleus-centrosome coupling[50], it is conceivable that DCX could also contribute to the recruitment of centrosomal DDR complexes in some of the identified neurons.

In addition to its relevance for memory, TLR9 activity was essential for the maintenance of neuronal genomic integrity. Knockdown of TLR9 inflammatory signalling showed that even mild stimuli, such as spontaneous or memory-related neuronal activity (and possibly other endogenous factors such as commensal microbes[51]) can trigger substantial genomic instability in the absence of TLR9-controlled DNA sensing and DDR. This suggests that the consequences of TLR9-based anti-inflammatory treatment strategies[52], although beneficial for restricting astrocytic and microglial activation, may prove detrimental for neuronal health if they compromise the functioning of the neuronal TLR9 pathway. Given that genomic instability is considered to be a gateway to accelerated senescence and psychiatric and neurodegenerative disorders[42,53,54], neuron-specific TLR9 and downstream RELA signalling are likely to emerge as promising preventive and therapeutic targets for preserving neurocognitive health.

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

## Methods

### Animal husbandry

Eight-week-old male and female C57BL/6 N mice were obtained from Envigo. *Tlr9*[fl/fl] (C57BL/6J-*Tlr9*[em1Ldm]/J) mice with *loxP* sites flanking exon 1 of the *Tlr9*, *Ifnar1*[fl/fl] (B6(Cg)-*Ifnar1*[tm1.1Ees]/J) mice with *loxP* sites flanking exon 3 of the type-1 interferon α/β receptor gene, *Rela*[fl/fl] (B6.129S1-*Rela*[tm1Ukl]/J) mice with *loxP* sites flanking exon 1 of the *Rela* gene, and *Sting1*-knockout mice C57BL/6J-Sting1gt/J; Goldenticket (Tmem173gt) mice were obtained from Jackson Laboratory and bred in institutional facilities. All mice were eight weeks of age at the beginning of experiments, unless otherwise specified. All mice were group housed (12 h:12 h light:dark cycle with lights on at 07:00, temperature 20–22 °C, humidity 30–60%) with ad libitum access to food and water (mice were switched to single housing one week before experiments). All experimental groups were mixed sex, consisting of approximately equal numbers of males and females. All procedures were approved by Northwestern University's Animal Care and Use Committee (protocols IS00002463 and IS00003359) and Albert Einstein's Animal Care and Use Committee (protocols 00001289 and 00001268) in compliance with US National Institutes of Health standards, and at Aarhus University in compliance with the Danish National Animal Experiment Committee (protocol 2021-15-0201-00801).

### Tissue collection

For all analyses, mice were euthanized by cervical dislocation, dorsal hippocampi immediately dissected and frozen in liquid nitrogen. Frozen tissue was stored at −80 °C until protein or RNA extractions were performed.

### Bulk RNA-seq

Read quality was assessed using FastQC[46] (v0.10.1) to identify sequencing cycles with low average quality, adapter contamination, or repetitive sequences from PCR amplification. Alignment quality was analysed using SAMtools flagstat with default parameters. Data quality was visually inspected as described previously[55]. Furthermore, we assessed whether samples were sequenced deep enough by analysing the average per base coverage and the saturation correlation for all samples using the MEDIPS R package[56]. The saturation function splits each library in fractions of the initial number of reads (ten subsets of equal size) and plots the convergence. The correlation between biological replicates was evaluated using Pearson correlation (function MEDIPS. correlation). Only data passing all quality standards were used for further analyses. Data were aligned to the genome using gapped alignment as RNA transcripts are subject to splicing and reads might therefore span two distant exons. Reads were aligned to the whole *Mus musculus* mm10 genome using STAR aligner58 (2.3.0e_r291) with default options, generating mapping files (BAM format). Reads were aligned to mouse genome *M. musculus* mm10 and counted using FeaturesCount as described previously[55]. Genes with significantly different fold change (false discovery rate-corrected $P < 0.05$) were classified as up-regulated or down-regulated.

### Isolation of nuclei and FACS

Fresh dorsal hippocampal tissue was collected from mice 96 h after CFC. Nuclei were isolated following an established protocol[57]. RNase inhibitor was added to the 6× homogenization buffer stable master mix at a final concentration of 1.2 U μl⁻¹. Tissue was homogenized using a Polytron homogenizer for 1.5 min. Samples were resuspended with the 50% iodixanol solution (50% iodixanol in 1× homogenization buffer) via gentle pipetting to make a final concentration of 25% iodixanol. Three millilitres of a 35% iodixanol solution (35% iodixanol in 1× homogenization containing 480 mM sucrose) was added to a new 15 ml Falcon tube. Three millilitres of a 29% iodixanol solution (29% iodixanol in 1× homogenization buffer containing 480 mM sucrose) was layered

above the 35% iodixanol mixture. A 4 ml 25% iodixanol solution was layered on the 29% solution. In a swinging-bucket centrifuge, nuclei were centrifuged for 20 min at 3,000*g*. After centrifugation, the nuclei were present at the interface of the 29% and 35% iodixanol solutions. This band with the nuclei was collected in a 300 μl volume and transferred to a pre-chilled tube. Following isolation, nuclei were counted and resuspended in fluorescence-activated cell sorting (FACS) buffer (1% BSA, 1 mM EDTA, and 0.2 U μl⁻¹ RNase inhibitor in PBS) at $2 \times 10^6$ nuclei per ml. A subset of nuclei was sorted by GFP expression on a high-speed cell sorter flow cytometer (Beckman Coulter MoFlowXDP cell sorter). See Supplementary Fig. 1 for the detailed gating strategy.

### snRNA-seq

snRNA-seq libraries were generated from up to 10,000 individual cells captured in an oil emulsion on a Chromium Controller (10x Genomics). cDNA was generated in the individual cell–gel bead emulsion micro-reactors while adding barcodes at the cellular and molecular level using the Chromium Next GEM Single Cell 3′ Kit v3.1(10x Genomics kit 1000268). The barcoded cDNAs from the individual cells were combined for the remaining library process. The unique molecular barcodes (UMIs) prevented amplification artefacts from skewing the analysis. The libraries were analysed on a Fragment Analyzer 5200 (Agilent Technologies) to ensure a normal size distribution with an average size of 450 bp and sequenced on an Ilumina Sequencer with the following read lengths: 28 bp for read 1, 10 bp for i7 index, 10 bp for i5 index and 90 bp for read 2 at a read depth of 20,000 reads per cell. Five libraries were generated and analysed (Supplementary Table 6).

### snRNA-seq analysis

The sequencing files in FASTQ format of each sample were aligned against mouse mm10 genome v4.0.0 and converted to gene count matrices using Cellranger software v7.0.1. Quality control and downstream analysis was performed using Seurat R package v4.3.0. Doublets were detected and removed using R package scDblFinder v1.13.13. Ambient RNA was detected and corrected using R package SoupX v1.6.2. Cells with less than 1000 detected genes, more than 4,000 detected genes, or more than 5% mitochondrial genes were excluded from further analysis. Samples were normalized using Seurat SCTransform function and then integrated and clustered using Seurat functions. Differential gene expression analysis was performed to compare cells from different samples in each cluster. Genes with more than 1.5-fold change and an adjusted $P$ value of less than 0.01 were defined as significant. Co-expression of two genes was assessed by comparing the cosine similarity of the target two genes and random chosen pair of genes. Gene set enrichment analysis was run on each cluster comparing samples with different condition against the Gene Ontology database using R package fgsea v1.20.0. Pathways with less than 0.05 adjusted $P$ value are considered significantly enriched. The pathway analysis was performed using the Reactome Patway database (version 86)[58]. Well-established markers for brain and blood-derived cell populations were used to define individual clusters[59,60].

### Gene ontology and interaction network analyses of bulk RNA-seq data

Functional protein association network analysis was performed using Search Tool for the Retrieval of Interacting Genes/Proteins (STRING) database (version 11.0)[61] for up-regulated genes. The network was processed applying kmeans clustering, setting cluster number to 4. Inflammatory response-associated genes were identified using Mouse Genome Database (MGI) Batch Query Genome Analysis Tool (v6.23). Selection criteria was classification within the GO term 'Inflammatory response' (ID: GO:0006954) using IMP (Inferred from mutant phenotype), IEA (Inferred from electronic annotation) and IBA (Inferred from biological aspect of ancestor) evidence codes. Functional classification of up-regulated genes was performed using PANTHER Classification

System engine, version 17.0. Genes were analysed through PANTHER Overrepresentation (test type: Fisher; correction: FDR; functional classification: PANTHER GO-Slim Biological Process, Mus musculus - REFLIST (21997)).

### RT² array

Differential expression of inflammatory response-associated genes between recent and remote memory was analysed using The Mouse Innate and Adaptive Immune Responses RT² Profiler PCR Array (Qiagen, 330231 PAMM-052ZA), according to manufacturer's instructions. Total RNA was extracted from dorsal hippocampi using PureLink RNA Mini Kit (ThermoFisher, 12183018 A). Tissue was dissected either 96 h ($n = 5$, pooled) or 21 days ($n = 5$, pooled) after CFC, resuspended in lysis buffer, flash frozen in liquid nitrogen and stored at −80 °C. RNA was extracted following manufacturer's instructions. cDNA was synthetized using the RT² First Strand Kit (Qiagen, 330401). For each experimental group, 500 µg of total RNA was used for the cDNA synthesis. Genomic DNA elimination reaction was prepared by mixing RNA with 2 µl of buffer GE in 10 µl total reaction volume. Real-time PCR reaction was performed using RT² SYBR Green qPCR Mastermix in duplicates for each sample (recent, remote). The PCR reaction was performed in an Applied Biosystems 7300 Real-Time PCR System with the following cycling conditions: 95 °C, 10 min; then 40 cycles of 95 °C, 15 s; 60 °C, 1 min. $\Delta C_t$ values for each gene were calculated by subtracting the average $C_t$ value of 6 housekeeping genes from the $C_t$ value of each target gene. Differential expression between recent and remote memory was calculated from $\Delta\Delta C_t$ value.

### Quantitative PCR analysis

Total RNA was extracted using RNeasy Plus Mini Kit (Qiagen, 74136). Reverse transcription was performed on 100 ng of total RNA Prime-Script RT Reagent Kit (Takara RR037A). Real-time PCR analysis was performed on a QuantStudio 6 Flex instrument (ThermoFisher) using SYBR green detection system (Applied Biosystems, 4367659) and primers specific for *Tlr9* (330001/PPM04221A-200), *Tlr7* (330001/PPM04208A-200) and *Tlr13* (330001/PPM41490A-200) (all from Qiagen). Housekeeping genes *Gapdh* and *Hprt* were used for normalization. $\Delta C_t$ for target genes was calculated using average $C_t$ of the two housekeeping genes. mRNA amount for each experimental group was expressed relative to naive group and calculated as $2^{\Delta C_t - \Delta C_{t_{naive}}}$.

### Cytosolic DNA extraction and analysis

Cytosolic DNA extraction was performed as previously described[62]. DNA was extracted using Mitochondria/Cytosol Fractionation Kit (Abcam, ab65320) from fresh tissue (dorsal hippocampus) and cleared extracts were treated with 1 mg ml⁻¹ proteinase K for 1 h at 55 °C, extracted with phenol:chloroform, treated with RNase A (1 mg ml⁻¹) for 1.5 h at 37 °C, and sequentially extracted with phenol:chloroform and chloroform. Purified cytosolic DNA was tested for nuclear DNA contamination by performing a 40-cycle PCR reaction using primers for vGlut1 (also known as *Slc17a7*) (forward: GTGGAAGTCCTGGAAACTGC, reverse: ATGAGCGAGGAGAATGTGG). For cloning of dsDNA, samples were treated with DNA polymerase I, large (Klenow) fragment (1 U µg⁻¹ DNA; NEB), supplemented with 33 µM of each dNTP, for 15 min. DNA was precipitated with sodium acetate/ethanol as described above, and dissolved in water. DNA samples were treated with Taq polymerase (NEB) for 20 min, and immediately cloned into pCR4-TOPO vector (Invitrogen), according to manufacturer's instructions. One Shot competent cells were transformed by adding 2 µl of the TOPO Cloning reaction into a vial of One Shot chemically competent *Escherichia coli*. Cells were incubated on ice for 30 min, heat-shocked for 30 s at 42 °C without shaking, and immediately transferred to ice. After 5 min, 250 µl of room temperature SOC medium was added, and tubes were placed horizontally in a shaker (200 rpm) at 37 °C for 1 h. Cells were pelleted at 6,000 rpm for 10 min, resuspended in 50 µl of SOC medium,

and spread on a pre-warmed selective plate. Plates were incubated at 37 °C overnight. All colonies from each plate were picked and placed into individual wells of a 96-well plate containing 50 µl of PBS. The sequence of cloned DNA fragments was determined by direct colony sequencing (ACGT).

### Primary cultures, treatment and live imaging

The hippocampi from post-natal day 0 (P1) C57BL/6 N male and female mice were isolated, and dissociated, as described previously[63]. Cells were plated in a 14-mm-diameter glass dish (MatTek, P35G-1.5-14-C) coated with poly-D-lysine (Sigma-Aldrich) at a density of 50,000 cells per cm² and grown in neuronal medium (Neurobasal Medium containing 1 mM GlutaMax, and 2% B27, all from ThermoFisher). Neurons were cultured for 14 days in vitro before treatments. Cells were treated with 25 µM *N*-methyl-D-aspartate (NMDA) for 10 min in fresh neuronal medium, washed twice, and incubated for 1 h with PicoGreen (dsDNA dye, 1:20,000), CellMAsk (cell membrane dye, 1:1,000), and MitoTracker (mitochondria dye, 1:20,000) diluted in neuronal medium. Cells were imaged on Nikon W1 spinning disc confocal microscope (Center for Advanced Microscopy, Northwestern University), 1 frame per 10 s at 100× magnification.

### Immunohistochemistry

Mice were anaesthetized with an intraperitoneal injection of 240 mg kg⁻¹ Avertin and transcardially perfused with ice-cold 4% paraformaldehyde in phosphate buffer (pH 7.4, 150 ml per mouse). Brains were removed and post-fixed for 24 h in the same fixative and then immersed for 24 h each in 20% and 30% sucrose in phosphate buffer. Brains were frozen and 50-µm sections were cut for use in free-floating immunohisto-chemistry[17] with the primary and secondary antibodies listed in Supplementary Tables 7 and 8, respectively. In addition to manufacturers' validation, all primary antibodies were validated by comparison to no primary control samples.

### PNN imaging

PNNs were visualized using *Wisteria floribunda* lectin (WFA) staining, a widely used approach for PNN visualization[40]. WFA staining was performed according to the manufacturer's instructions. In brief, endogenous peroxidase was inactivated with hydrogen peroxide. Following streptavidin/biotin and Carbo-Free blocking, sections were incubated with biotinylated WFA (Vector Biolaboratories), Vectastain ABC system, and fluorescein, coumarin or rhodamine isothiocyanate (Akoya Biosciences). Sections were mounted using FluorSave (Millipore-Sigma).

### Microscopy, image analysis and quantification

Low-magnification images (up to 60× magnification) were captured with a Leica microscope with a Leica DFC450 C digital camera whereas high-magnification images and *z*-stacks (60–100× magnification) were captured using a confocal laser-scanning microscope (Olympus Fluoview FV10i). All quantifications were performed with ImageJ. Clusters of γH2AX-positive neurons were first identified, and analyses were performed in the surrounding (~100 µm × 100 µm) area. Thus, the numbers are representative of regions of interest rather than average of the CA1 subfield. For time-course analyses, we counted γH2AX neurons in 180 neurons per mouse (3 consecutive slices of 60 neurons). With 6 mice per group, this amounted to 1,000 neurons per time point. In most of the other molecular targets we counted 60 neurons per mouse using a 60–100× objective. All images were converted to binary format, and for each cell showing γH2AX we determined the background and applied a threshold twice above the background signal. We thus obtained similar results across different antibodies and conditions. All analyses were performed with Fiji/ImageJ. The JACoP Plugin for object-based co-localization was used to determine co-localization by comparing the position of the centroids of the nuclei of the colour channels. Their respective coordinates were then used to define

structures separated by distances equal to or below the optical resolution[45]. Volume and 3D viewer Plugins were used for 3D reconstruction of z-stacks, whereas plot profile and surface plot functions were used for analyses of clusters at lower (40×) magnification. Coloc2 was used to determine correlation of expression levels of different fluorophore signals. The analyse particles, plot profile and measure functions were used to determine the number, size, distribution and distance between indicated signals.

## Fluorescent multiplex v2 RNAscope

Naive mice (n = 4) or mice subjected to CFC (n = 4) were perfused 96 h later with ice-cold 0.1 M PBS and 4% paraformaldehyde in PBS and processed as described above. RNAscope was performed according to the manufacturer instructions. To visualize Hsp90b1 and Dcx mRNA and NeuN protein RNAscope Multiplex Fluorescent Reagent Kit v2 (ACD Biotechne, 323100) and RNA–Protein Co-Detection Ancillary kit (ACD Biotechne, 323180) were used. In brief, slides were dried at 60 °C in an oven for 30 min, dehydrated in ethanol, treated with hydrogen peroxide for 10 min at room temperature, washed in water and boiled in co-detection target retrieval reagent (around 98 °C) for 5 min. The sections were incubated with anti-Neun antibody (1:500, Sigma, ABN78), fixed with 10% Neutral Buffered Formalin (VWR, GEN0786-1056), protease plus, and rinsed with sterile water. The hybridization step was performed by incubating the sections with the following probes: Mm-Hsp90b1-C1 (ACD Biotechne, 556051), Mm-DCX-C2 (ACD Biotechne, 478671-C2), Mm-PPIB (positive control probe, ACD Biotechne, 320881) and dabB (negative control probe, ACD Biotechne, 320871) for 2 h at 40 °C and stored overnight in 5× saline sodium citrate. The hybridization was amplified with AMP 1 and AMP 2. All amplification and development were performed at 40 °C, and 2 × 2 min of washes in ACD wash buffer was performed after each step. For C1 probe TSA Vivid Fluorophore Kit 520 (Biotechne, 7523) was used, for C2 and C3, positive and negative probes TSA Vivid Fluorophore Kit 650 (7527) was used. Sections were incubated with goat anti-rabbit Alexa-568 secondary antibody (1:300, Invitrogen, A11036) and DAPI solution (ACD bio), and mounted with Prolong Gold Antifade Mountant (ThermoFisher Scientific, 33342). Images were acquired with an Andor BC43 spinning disk confocal microscope (Oxford Instruments) controlled by the Fusion software from Andor, using a 10× air 0.45 NA objective, a 60× oil immersion 1.42 NA objective, a CMOS camera (6.5 µm pixel; 2,048 × 2,000 pixels generating 16-bit, monochrome images) with no binning. Samples were illuminated with 4 fixed wavelengths of 405 nm, 488 nm, 561 nm and 638 nm. Dorsal hippocampus overview was obtained with 3 × 3 stitching (10% overlap) using the 10× objective and the region of interest (CA1 pyramidal layer, closest to the midline) was afterward imaged at higher magnification using the 60× objective. Image analysis was performed after thresholding using the analyse particle function in ImageJ using four slices per mouse. The number of particles was analysed per 60 neurons per slice (240 neurons per mouse).

## Fear conditioning

CFC was performed in an automated system (TSE Systems) as previously described[29]. In brief, mice were exposed for 3 min to a novel context, followed by a foot shock (2 s, 0.7 mA, constant current). TFC was performed by exposing the mice to for 3 min to a novel context, followed by a 30 s tone (75 dB SPL, 10 kHz, 200 ms pulse), a 15 s trace, and a foot shock (2 s, 0.7 mA, constant current)[63]. DFC was performed as described for TFC, except that trace was omitted so that shock immediately followed the end of the tone. At indicated time points, mice were tested for memory retrieval by re-exposing them to the same context (context test), or to a tone presented over 30 s in a different context (tone test after TFC or DFC). Testing consisted of 3 min in the conditioning context, during which freezing was measured every 10 s. Freezing was expressed as a percentage of the total number of observations during which the mice were motionless. Activity was recorded automatically by an infrared beam system and expressed in cm s$^{-1}$. The individual experiments with wild-type mice were not performed on littermates, so we did not apply randomization procedures, but with all genetic lines bred in our facility, littermates were randomly assigned to different experimental groups to minimize litter effects. The behavioural tests were performed blindly, either by experimentalists who were unaware of the treatments because the solutions were coded or by experimenters unaware of the experimental design. The experimenter performing the tests was not aware of the numbering code.

## Surgery and cannulation

Double-guided cannulas (Plastic One) were implanted in the dorsal hippocampus as described previously[17]. Mice were anesthetized with 1.2% tribromoethanol (vol/vol, Avertin) and implanted with bilateral 26-gauge cannulas using a stereotaxic apparatus (Kopf, model 1900). Stereotaxic coordinates for the dorsal hippocampus were 1.8 mm posterior, ±1.0 mm lateral and 2.0 mm ventral to bregma.

## Pharmacological treatments

All oligonucleotides and drugs were injected into the dorsal hippocampus at a volume of 0.25 µl per side, at a rate of 0.15 µl min$^{-1}$ using micro-infusion pumps (Model UMP3T-1 UltraMicroPump 3 with SMARTouch Controller). ODN2088 was injected at doses of 125 and 300 ng per mouse corresponding to 4 and 8 nmol, respectively. Based on pilot experiments, the cGAS and STING antagonists were injected at doses of 10 and 50 ng in 250 nl per mouse.

DNase I (Sigma-Aldrich D4513, lot SLCQ3662, diluted in water) was injected intraperitoneally with 50 U DNase I in 200 µl saline, 24 h and 12 h before and 1 h after CFC, or intrahippocampally with 50 U DNase I per mouse at volume of 0.5 µl per side, at a rate of 1 µl min$^{-1}$ 24 h before and 1 h after test. DNase I and S1 nuclease (Thermo Scientific, EN0321) treatment were injected into the dorsal hippocampus at dosage 5 U DNase I + 10 U S1 in 20% glycerol/artificial cerebrospinal fluid, vehicle: 20% glycerol/artificial cerebrospinal fluid 4 h before the memory test. Treatment schedules and doses were designed based on published data[64] and including both high and low doses as well as pre-CFC and pre-test injections.

For depletion of microglia, Pexidartinib (PLX-3397, HY-16749 lot: 212013 MedChemExpress) was formulated into 5053 PicoLab Rodent Diet 20 at the concentration of 290 ppm (W.F. Fisher and Son). The mice were fed for four weeks before the test. LabDiet 5053 served as a control diet. At the end of experiments, all brains were collected for histological determination of cannula placements or immunohistochemistry.

## Virus injections

All viruses (listed in Supplementary Table 9) were injected at a volume of 0.5 µl per side, 1.8 mm posterior, ±1.0 mm lateral and 2.0 mm ventral to bregma, at a rate of 0.15 µl min$^{-1}$. At the time of virus infusion, mice were eight weeks old. CFC and memory tests were performed from weeks 13 to 17. One day after the completion of behavioural testing, mice were intracardially perfused, and all brains were collected, and virus spread was confirmed by immunohistochemical analysis of GFP or mCherry.

## Labelling CFC-activated CA1 neurons using robust activity marking

We labelled the CA1 neurons in the dorsal hippocampus activated during CFC using doxycycline-off (off-Dox) activity-dependent cell tagging with the Robust Activity RAM system[31] coupled to the human Fos minimal promoter with four tandem repeats of an enhancer module (PRAM) as recently described in detail[11]. In brief, we put two groups of wild-type mice on a doxycycline diet one day before injecting AAV2/9-PRAM:d2tTA-TRE:NLS-mKate2 and took them off the diet after 9 days, followed by CFC the day after. Mice were left undisturbed in our

testing facility without doxycycline for additional two days. Following two days, mice were put back on the doxycycline diet and 96 h after CFC, one group was subjected to memory reactivation (context test) and euthanized 1 h later. An additional, non-reactivated group served as control. This later group was only used internally to ensure that there was no Fos response without memory reactivation.

## Statistics and reproducibility

Statistical power to detect anticipated effect sizes was determined using power analysis (calculator at http://www.stat.ubc.ca/~rollin/stats/ssize/n2.html) conducted on representative samples of previous work and pilot experiments. For all proposed experiments, minimum power is set at 0.90 to detect an $\alpha = 0.05$ (two-sided test) for a difference in means from 20% to 40%, with a 15% common standard deviation. To prevent litter effects, mice from the same litter were assigned to different experimental groups. Viruses were injected by experimenters aware of the construct but the mice were then assigned coded numbers by the laboratory technician. The code was available after quantification and before analyses. Statistical analyses were performed using GraphPad Prism. Mice with misplaced virus infusion or cannulas, and mice with less than 70% viral expression in the CA1 were excluded. One-way ANOVA followed by Tukey's test was used for post hoc comparisons of three or more experimental groups (only when ANOVA was significant) whereas Student's $t$-test was used for comparison of two experimental groups. Homogeneity of variance was confirmed with Levene's test for equality of variances. On indicated data, we performed correlation analyses and report Pearson's $r$ coefficients. Significant changes of co-localization or activation (%) were determined using the Chi-square test. A priori determined post hoc analyses following Chi-square tests were performed using Bonferroni-corrected alpha levels as the original overall alpha level ($\alpha = 0.05$) was divided by the number of tests being conducted. These adjusted alpha levels were used as the new significance threshold for each individual test. All comparisons were conducted using two-tailed tests and the $P$ value for all cases was set to <0.05 for significant differences. Data are expressed as mean ± s.e.m. Statistically significant differences are indicated as $*P < 0.05$, $**P < 0.01$, $***P < 0.001$, $****P < 0.0001$ and $*****P < 0.00001$. All cellular and molecular effects were shown in a minimum of six biological replicates (Figs. 1e and 2e and Extended Data Figs. 3f and 9). Two experimental replicates were performed for all time-course, *Tlr9*-KO, *Rela*-KO, *Ifnar1*-KO and WT−*cre* experiments (Figs. 1c,d, 2a–d and 5a,c and Extended Data Figs. 2–4, 11 and 12). All significant behavioural effects were replicated at least three times using wild-type, virus-specific and cell-specific control groups. All main gene expression effects were replicated with four different approaches (bulk RNA-seq, quantitative PCR arrays, quantitative PCR and snRNA-seq). Replicates produced similar results relative to initial or representative experiments.

## Figures

Figures were created and edited using Adobe Illustrator CS6 (Adobe, v27.5, RRID: SCR_010279). Figs. 1a, 1b, 2a, 2e, 3a, 3c, 3d, 4a, and 5a, as well as Extended Data Figs. 4c, 4d, 5a, 5b, 5c, 5d and Supplementary Fig. 1 were created using BioRender.com.

## Ethics declaration

All animal procedures used in this study were approved by the Northwestern University IACUC, Albert Einstein Medical College IACUC and Danish National Animal Experiment Committee, and complied with federal regulations set forth by the National Institutes of Health.

## Reporting summary

Further information on research design is available in the Nature Portfolio Reporting Summary linked to this article.

## Data availability

The bulk RNA-seq data are available at the NCBI Gene Expression Omnibus under accession GSE174076. The snRNA-seq data are available at the NCBI Gene Expression Omnibus database under accession GSE254780. Source data are provided with this paper.

## Code availability

Source code is available at https://github.com/RadulovicLab/Nature-2024.

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

**Acknowledgements** This work was supported by NIH grants R01MH108837 and R01MH078064 to J.R., Lundbeck Foundation grant R310-2018-3611 to J.R., Lundbeck Foundation grant R307-2018-3667 to J.K., and DFG priority programme 1738, SFB1286, MBExC of Germany's Excellence Strategy—EXC 2067/1 390729940 to A.F. The authors thank Y. Lin for providing the PRAM virus; and the Flow Cytometry (especially J. Zhang and F. Aodengtuya), Genetics and Computational Genomics Cores at Albert Einstein College of Medicine. Illustrations of nonscientific data were created with BioRender.com through an institutional licence.

**Author contributions** V.J. performed quantitative PCR, cloning, in vivo pharmacological experiments and in vitro experiments with primary hippocampal cultures, and helped with the preparation of the manuscript. E.M.W. processed the nuclei for snRNA-seq, and with A. Cicvaric, performed the time-course studies for dsDNA break and RELA detection, behavioural and immunohistochemical analyses of knockdown effects, and helped with the preparation of the manuscript. H.Z., K.K.P., A. Cicvaric and T.E.B. performed the behavioural and histological analyses of TLR9 knockdown. Z.P. and A. Carboncino performed most of the immunohistochemical experiments and microglia diet studies. M.M., N.Y., H.L. and J.K. performed the experiments with STING knockouts. H.L. performed the RNAscope assay and imaging. A.F. and F.S. performed the bulk RNA-seq and, with V.J., bioinformatic analyses. X.Z. performed the bioinformatic analyses of the snRNA-seq data. J.R. designed the overall study, helped with imaging and data analysis, and wrote the manuscript.

**Competing interests** The authors declare no competing interests.

**Additional information**
**Correspondence and requests for materials** should be addressed to Jelena Radulovic.

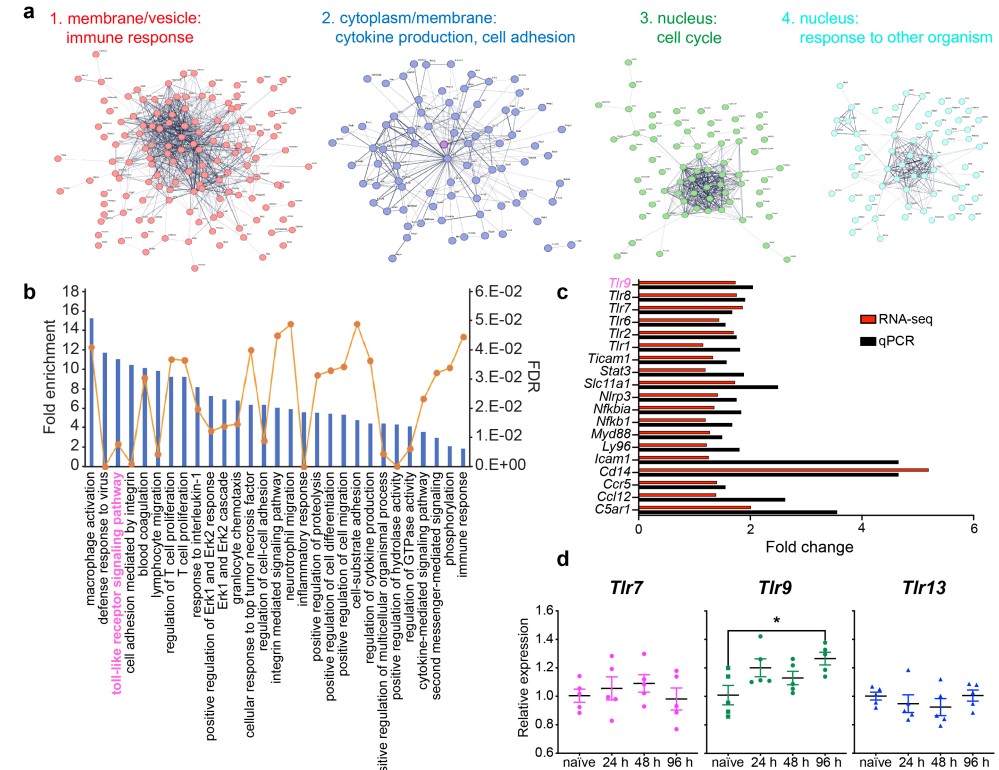

**Extended Data Fig. 1 | Up-regulated genes and pathways during memory formation. a** Expanded reactome analysis (String database, STRING-db.org) outlining individual functional clusters of up-regulated genes enriched in inflammation, cytokine release, and cell cycle/DNA repair pathways. **b** Gene ontology analysis showing enrichment of Tlr signaling. **c** Up-regulation of the Tlr9 pathway replicated in a separate set of hippocampi collected 96 h or 28 days after CFC with qPCR microarrays customized for immune response genes. **d** qPCR comparing the level of selected Tlr (Tlr7, Tlr9, and Tlr13) obtained 24, 48, or 96 h after CFC relative to naïve samples showing significant up-regulation of Tlr9 ($n$ = 5 mice/group; one way ANOVA Tlr7: p = 0.6677, $F_{(3,16)}$ = 0.5306; Tlr9: p = 0.0317, $F_{(3,16)}$ = 3.783; Tlr13: p = 0.5903, $F_{(3,16)}$ = 0.6569). Data represented as mean ± s.e.m., $^{ns}$p > 0.05; ****p < 0.0001; ***p < 0.001; **p < 0.01; *p < 0.05.

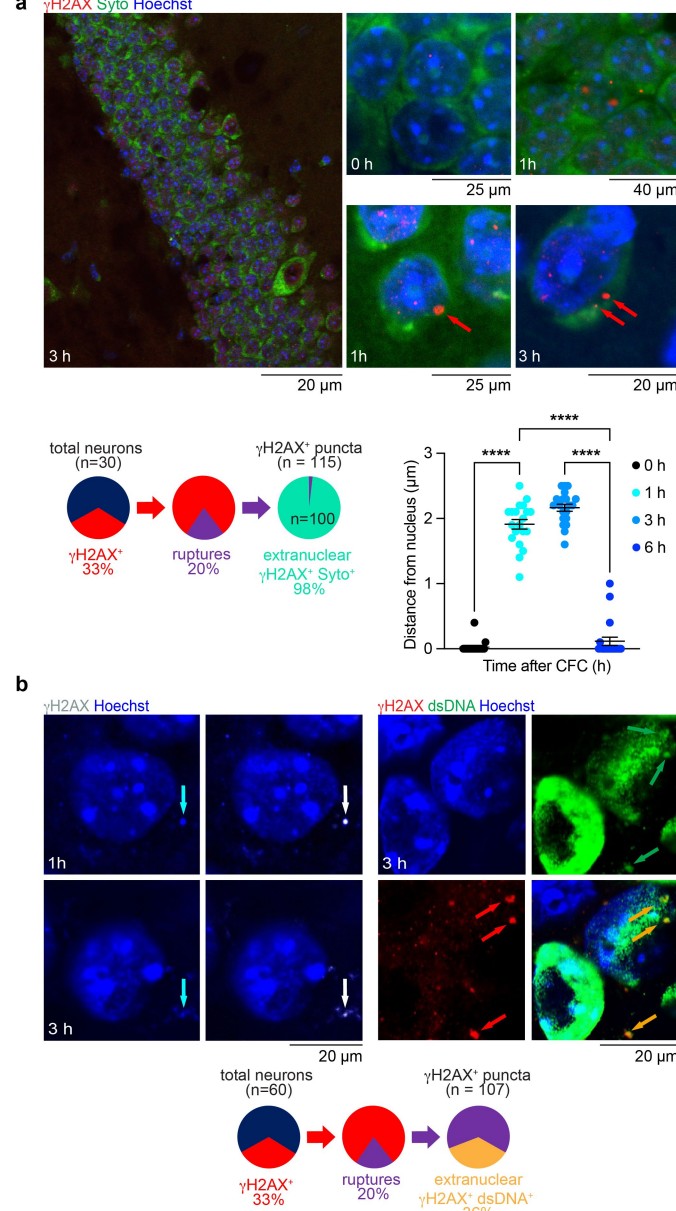

**Extended Data Fig. 2 | Detection of extranuclear γH2AX signals.**
**a** Perinuclear localization of extranuclear γH2AX signals 1-3 h post CFC in
RNA-rich compartments (Syto dye; $n = 115$ cells, 98%) including ER and
ribosomes, but not nucleoli. Measurements of the distance from nuclei
revealed that most signals were perinuclear, co-localizing with RNA-rich ER
and ribosomal compartments ($n = 80$ cells; one-way ANOVA $p < 0.0001$,
$F_{(3,76)} = 400.3$). **b** Co-localization of extranuclear dsDNA and γH2AX signals in
the vicinity of ruptured nuclei 1-3 h post CFC in 36% of the γH2AX-positive
puncta ($n = 107$ nuclei total). Data represented as mean ± s.e.m., [ns]$p > 0.05$;
****$p < 0.0001$; ***$p < 0.001$; **$p < 0.01$; *$p < 0.05$.

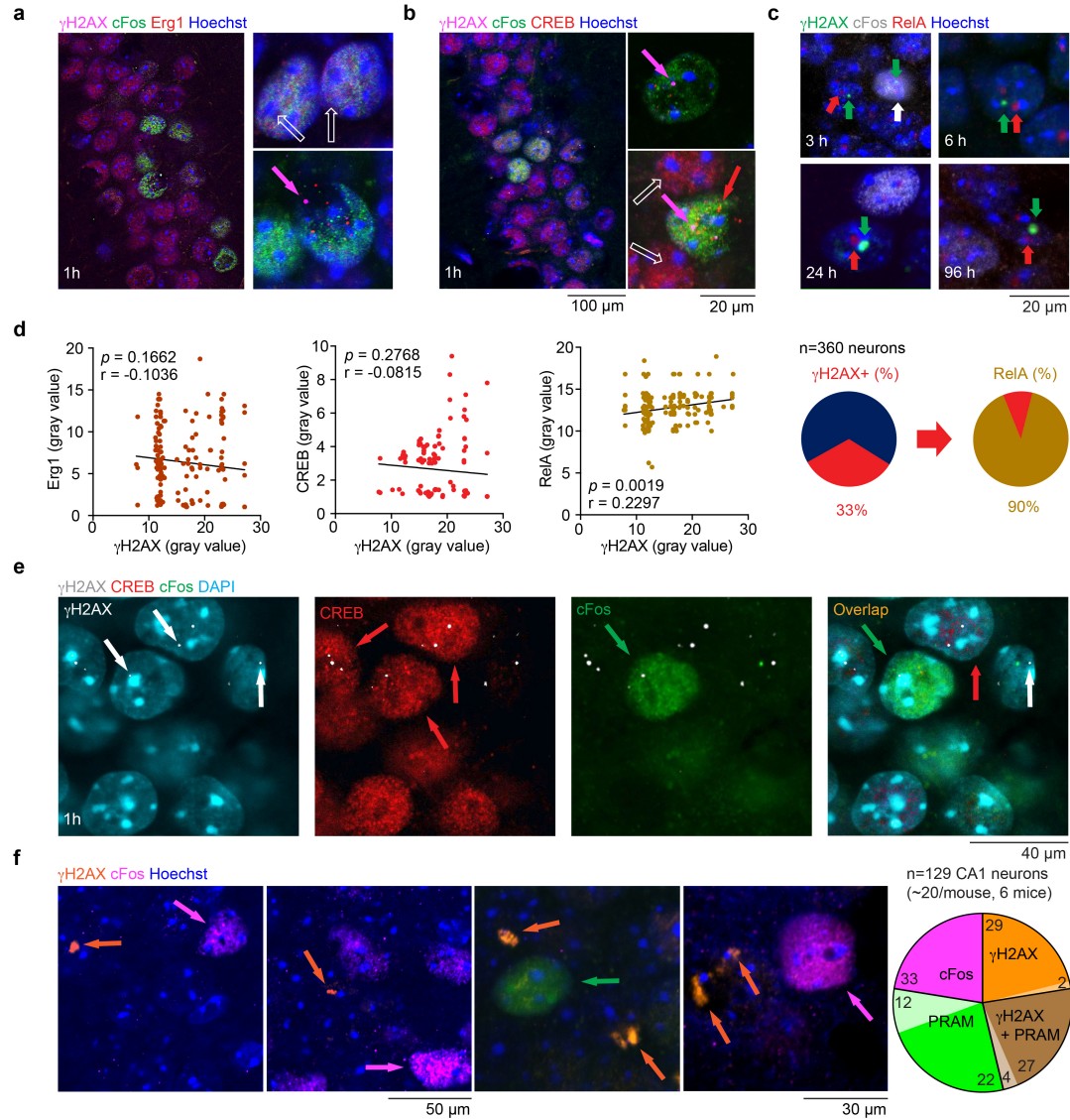

**Extended Data Fig. 3 | Relationship between the immediate early gene response, inflammatory response, and γH2AX signals. a, b, c** Representative micrographs of γH2AX and cFos signals with Erg1, CREB, and RelA, respectively. **d** Regression analyses of γH2AX signals with Erg1, CREB, and RelA, respectively in individual neurons. 1 h after CFC, γH2AX puncta show no correlation with Erg1 (*n* = 120 neurons, 20 neurons/mouse). Similarly, γH2AX signals did not correlate with CREB levels (*n* = 120 neurons, 20 neurons/mouse). Significant correlation and neuronal overlap of RelA and γH2AX-positive puncta (*n* = 120 neurons, 20 neurons/mouse) providing support for an inflammatory component of dsDNA damage and DDR. At time points of optimal RelA detection, cFos levels were expectedly low. The *r* and *P* values are depicted in the individual graphs. **e** Examples of random occurrence of γH2AX puncta relative to individual IEG. **f** Segregated γH2AX and cFos labeling 96 h after context memory reactivation.

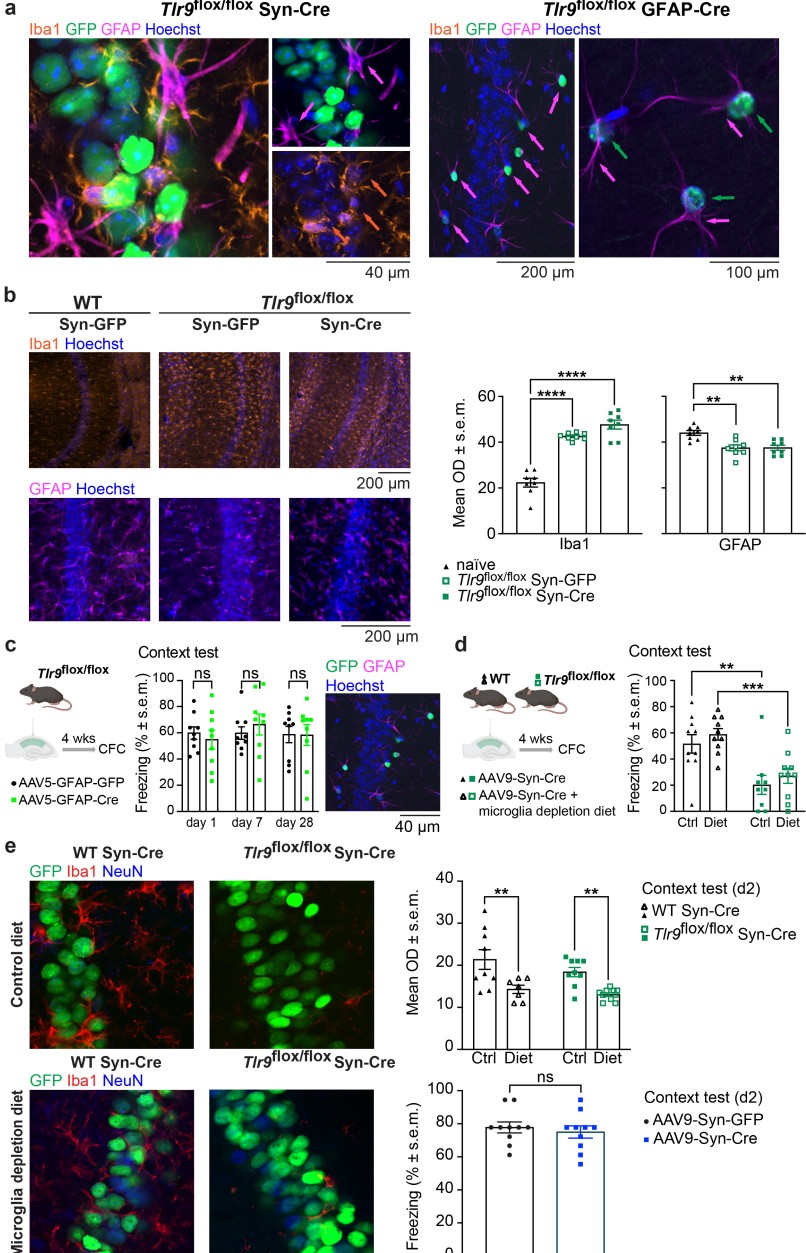

**Extended Data Fig. 4 | Responses of hippocampal astrocytes and microglia to viral infusions and diet. a** Immunohistochemistry demonstrating lack of overlap between GFP signals stemming from Syn-Cre expression and markers specific for astrocytes (GFAP) or microglia (Iba1, left). GFAP-driven Cre expression was only found in astrocytes (right). **b** Increased density of the microglial protein Iba1 (WT, Syn-GFP, and Syn-Cre, $n = 8$ mice/group, one-way ANOVA, $p < 0.0001$, $F_{(2,21)} = 69.83$) and decreased density of astrocytic protein GFAP ($n = 8$ mice/group, one-way ANOVA, $p = 0.0006$, $F_{(2,21)} = 10.65$) in mice injected with AAV9-Syn-Cre-GFP or control virus AAV9-Syn-GFP when compared to naïve WT controls. **c** Intact CFC after hippocampal astrocytic deletion of Tlr9 (left, middle) induced by intrahippocampal injection of astrocyte-specific GFAP-Cre in Tlr9$^{flox/flox}$ mice when compared to the control group injected with GFAP-GFP (GFAP-GFP and GFAP-Cre $n = 9$ mice/group, two-way ANOVA RM, Factor: Virus, $p = 0.9560$, $F_{(1,16)} = 0.0031$, Factor: Test, $p = 0.5451$, $F_{(1.716,27.46)} = 0.572$, Virus × Test, $p = 0.5928$, $F_{(2,32)} = 0.5315$). Micrograph showing viral expression in astrocytes (right, labeled by GFAP). **d** WT and Tlr9$^{flox/flox}$ mice injected

intrahippocampally with neuron-specific Syn-Cre and fed for 4 weeks with regular or microglial depletion diet prior to CFC (WT control and diet, Tlr9 KO control and diet $n = 9$-10 mice/group). While depleting microglia, the diet did not affect the Tlr9KO induced freezing impairments (two-way ANOVA, Factor: Genotype, $p < 0.001$, $F_{(1,35)} = 25.94$, Factor: Diet, $p = 0.2611$, $F_{(1,35)} = 1.305$, Genotype × Diet, $p = 0.9731$, $F_{(1,35)} = 0.0012$). **e** Decreased density of the microglial protein Iba1 after a 4-week diet containing the CSF1R kinase inhibitor PLX3397 when compared to regular diet prior to CFC (left). Similar reduction was found in WT (control $n = 9$ mice, diet $n = 7$ mice) and Tlr9$^{flox/flox}$ mice (control $n = 8$ mice, diet $n = 9$ mice) injected with Syn-Cre (two-way ANOVA, Factor: Genotype, $p = 0.1487$, $F_{(1,31)} = 2.193$, Factor: Diet, $p = 0.0001$, $F_{(1,31)} = 19.06$, Genotype × Diet, $p = 0.5433$, $F_{(1,31)} = 0.3777$) (top right). Additional control experiment showing similar freezing of WT mice after intrahippocampal injection of Syn-Cre and Syn-GFP ($n = 10$ mice/group; two-tailed unpaired $t$ test, $t_{18} = 0.5571$, $p = 0.5843$, bottom right). Data represented as mean ± s.e.m., $^{ns}p > 0.05$; $^{****}p < 0.0001$; $^{***}p < 0.001$; $^{**}p < 0.01$; $^{*}p < 0.05$.

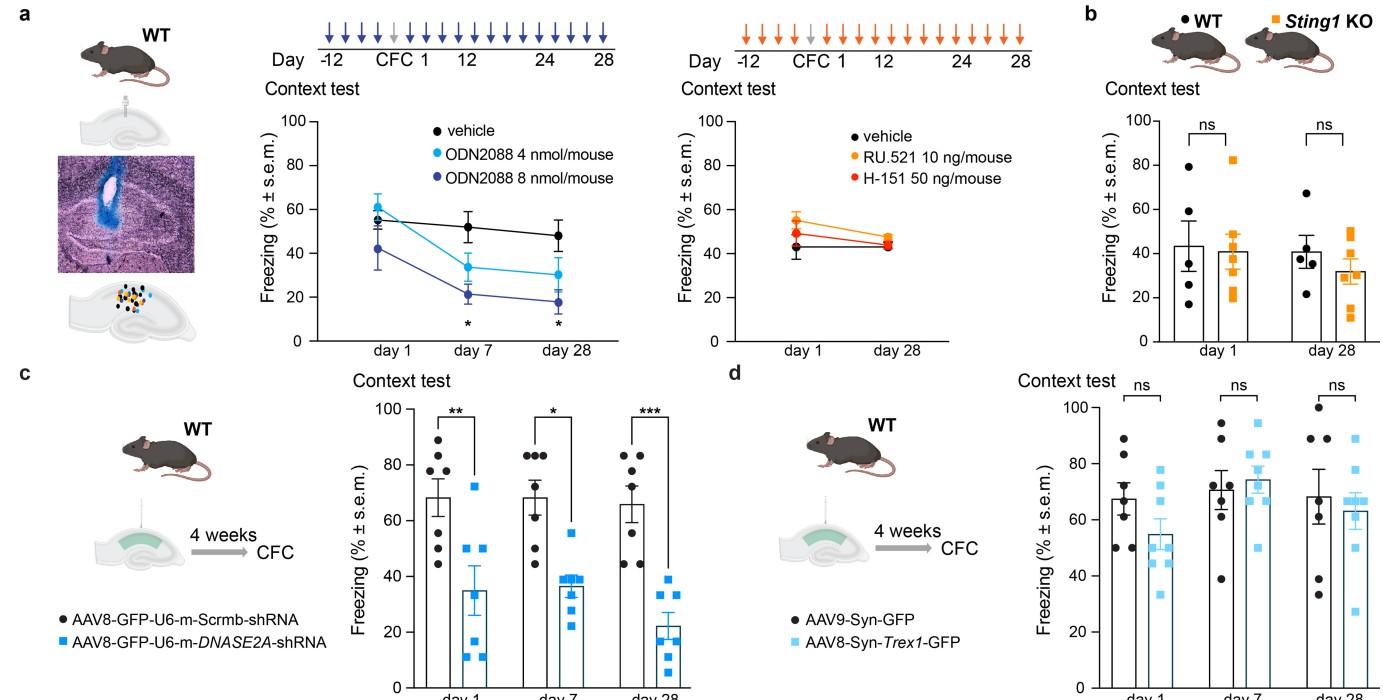

**Extended Data Fig. 5 | Effect of pharmacological manipulations of Tlr9 and GAS/STING pathways on CFC. a** Cannula localization and individual placements (left). Dose-dependent impairment of CFC by pharmacological inhibition of Tlr9 with ODN2088 (4 nmol/mouse $n = 8$; 8 nmol/mouse $n = 7$, vehicle $n = 17$; middle, two-way ANOVA RM, Factor: Drug, p = 0.0238, $F_{(2,29)} = 4.2630$, Factor: Test, p < 0.0001, $F_{(1.860,53.54)} = 12.66$, Drug × Test, p = 0.0748, $F_{(4,58)} = 2.249$), and lack of effect of the GAS and STING inhibitors RU-521 ($n = 8$ mice/group) and H-151 ($n = 8$ mice/group), respectively (right, two-way ANOVA RM, Factor: Drug, p = 0.1096, $F_{(2,34)} = 2.361$, Factor: Test, p = 0.1708, $F_{(1,34)} = 1.958$, Drug × Test, p = 0.6029, $F_{(2,34)} = 0.5136$). **b** Intact CFC in STING knockout mice ($n = 7$ mice/group) versus WT littermates ($n = 5$ mice/group; two-way ANOVA RM,

Factor: Genotype, p = 0.5132, $F_{(1,10)} = 0.4596$, Factor: Test, p = 0.4794, $F_{(1,10)} = 0.5398$, Genotype × Test, p = 0.7141, $F_{(1,10)} = 0.1421$). **c** Impaired CFC after dorso-hippocampal injection of DNAse2 shRNA ($n = 7$ mice/group) when compared to scrambled shRNA ($n = 7$ mice/group; two-way ANOVA, Factor: Virus, p < 0.0001, $F_{(1,36)} = 48.02$, Factor: Test, p = 0.3667, $F_{(2,36)} = 1.032$, Virus × Test, p = 0.6051, $F_{(2,36)} = 0.5094$). **d** Overexpression of TREX1 ($n = 8$ mice/group) did not affect CFC when compared to the GFP control ($n = 7$ mice/group; two-way ANOVA, Factor: Virus, p = 0.3918, $F_{(1,39)} = 0.7500$, Factor: Test, p = 0.2407, $F_{(2,39)} = 1.478$, Virus × Test, p = 0.4763, $F_{(2,39)} = 0.7559$). Data represented as mean ± s.e.m., [ns]p > 0.05; ****p < 0.0001; ***p < 0.001; **p < 0.01; *p < 0.05.

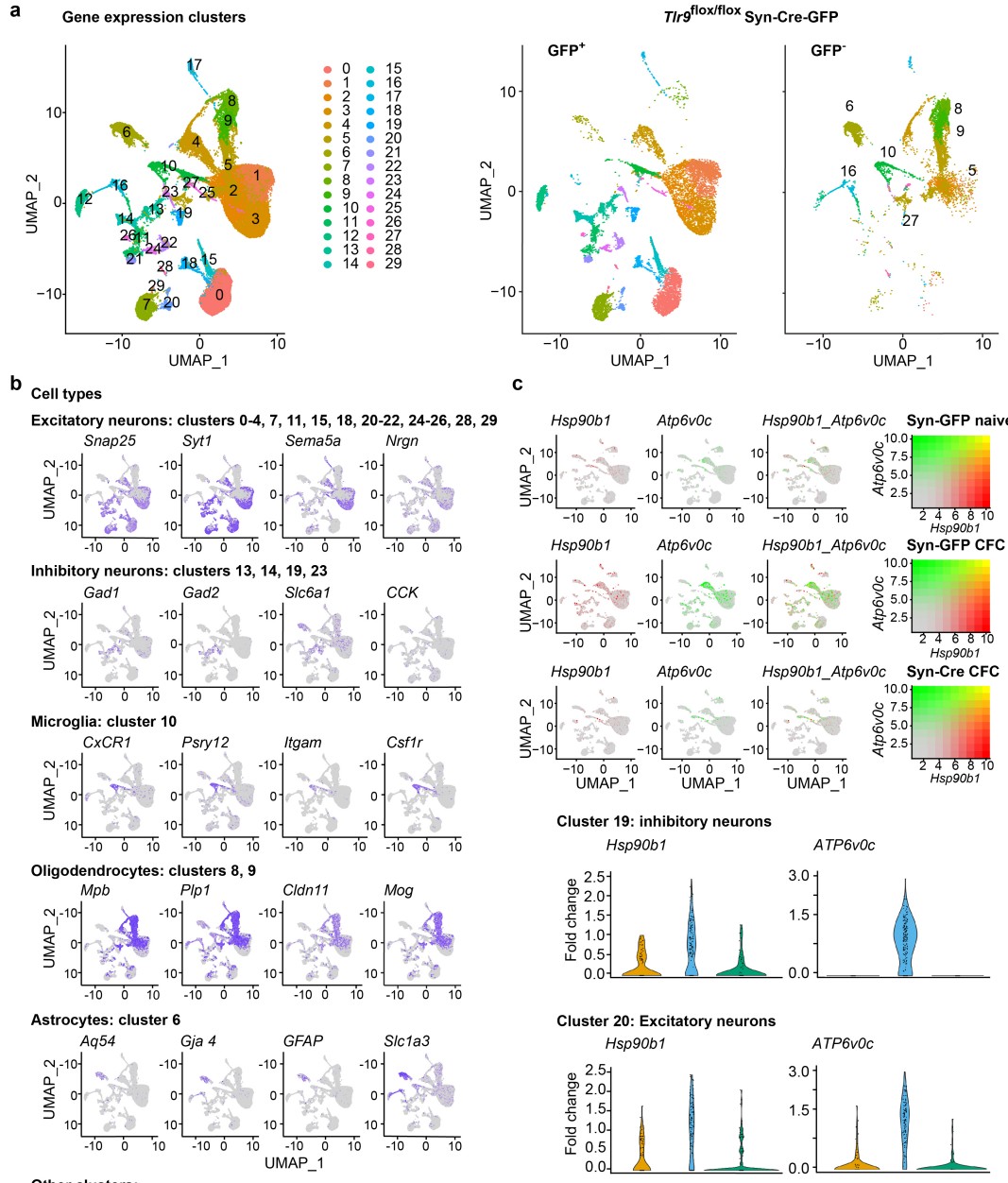

**Extended Data Fig. 6 | Bioinformatic analysis of snRNA-Seq data.**
**a** Unsupervised algorithm identifying 29 cell clusters (left). Nuclei obtained from hippocampi Tlr9$^{flox/flox}$ mice injected with Syn-Cre but not exposed to CFC were prepared as described in Fig. 4 but separated using fluorescence-activated cell sorting (FACS) and the separated populations of GFP$^+$ and GFP$^-$ nuclei were subjected to snRNA-Seq using the same approaches as described for unsorted samples. When compared to the total neuron clusters (left), the gene profiles of GFP$^+$ nuclei showed a pattern consistent with neuron-specific expression whereas the gene profiles of GFP$^-$ nuclei showed a pattern consistent with expression in non-neuronal cells (right). **b** Identification of main cell populations based on known markers for neuronal and non-neuronal cells. **c** Examination of cosine similarity of Atp6v0c and Hsp90b1 across clusters revealed a coefficient of 0.6 and a significant above chance association ($p < 0.01$), suggesting co-expression across cell types (top). Violin plots showing up-regulation of Atp6v0c and Hsp90b1 by CFC and down-regulation by Tlr9KO (bottom).

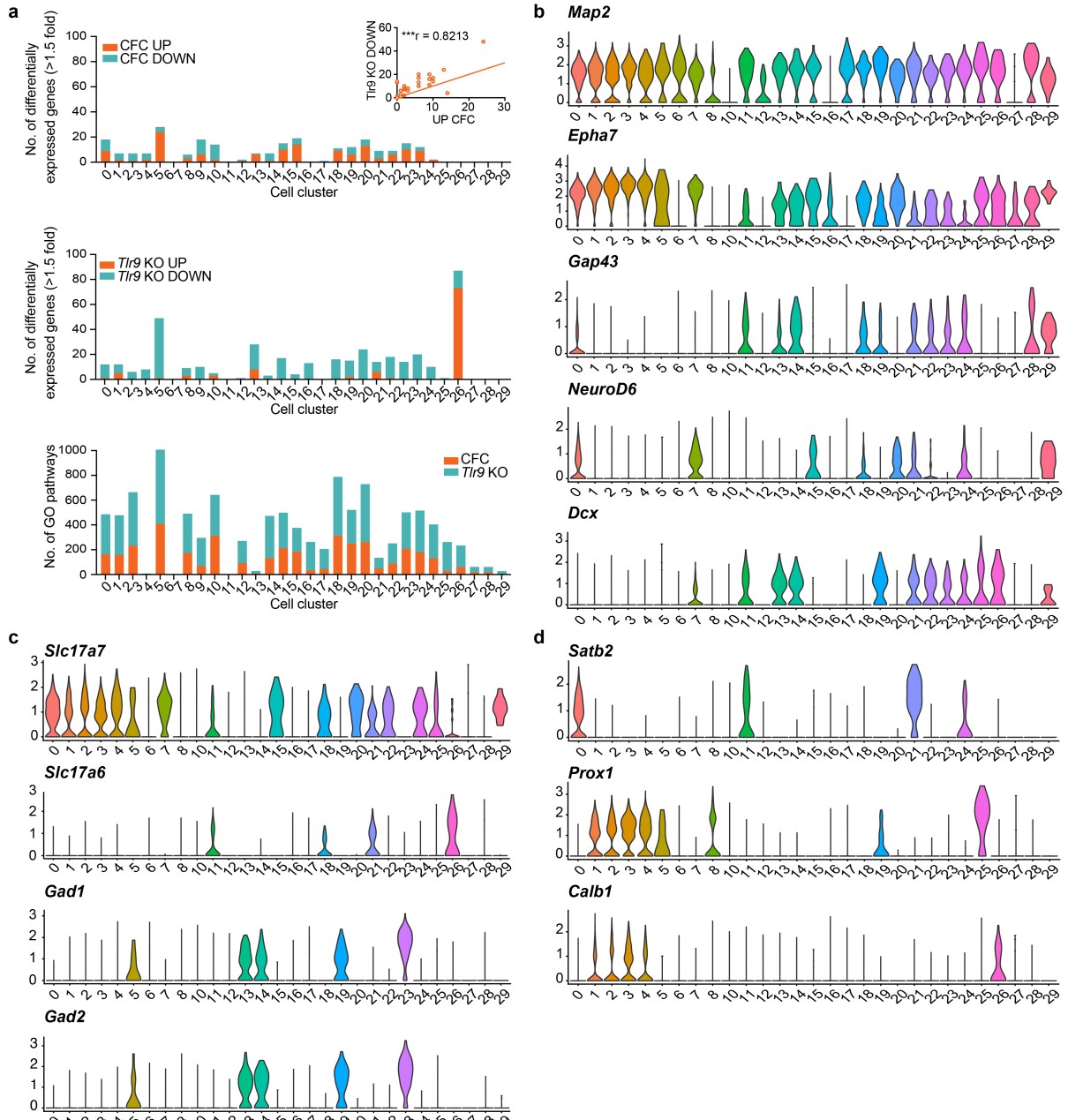

**Extended Data Fig. 7 | Cluster-specific gene expression and presence of neuronal markers. a** Up-regulation of gene expression after CFC (top) and down-regulation after Tlr9KO (middle) is predominant in neuronal clusters. A significant correlation was found between CFC increased vs *Tlr9* KO decreased changes of gene expression by cluster (top corner). A unique pattern opposite the effect of *Tlr9* KO is seen in cluster 26, showing a strong increase of gene expression, most of which were involved in control of axon guidance and cell-matrix adhesion. This cluster uniquely expresses Slc17a6 coding for vGlut2 (see **c**). Gene ontology analysis revealing similar patterns of up-regulation and down-regulation of functional pathways (bottom). **b** Neuronal phenotyping with conserved markers revealing presence of Dcx in several excitatory and all inhibitory neuronal clusters in addition to immature DGGC (cluster 25). **c** The expression of excitatory and inhibitory neuron markers as well as **d** CA and DGGC markers were consistent with the clustering method and identified clusters 0, 7, 11, 15, 18, 20, 21, 22, 24, 26, 28, and 29 as excitatory CA neurons, clusters 1-4 as DGGC, cluster 25 as immature DGGC, and clusters 13, 14, 19, and 23 as inhibitory neurons.

a

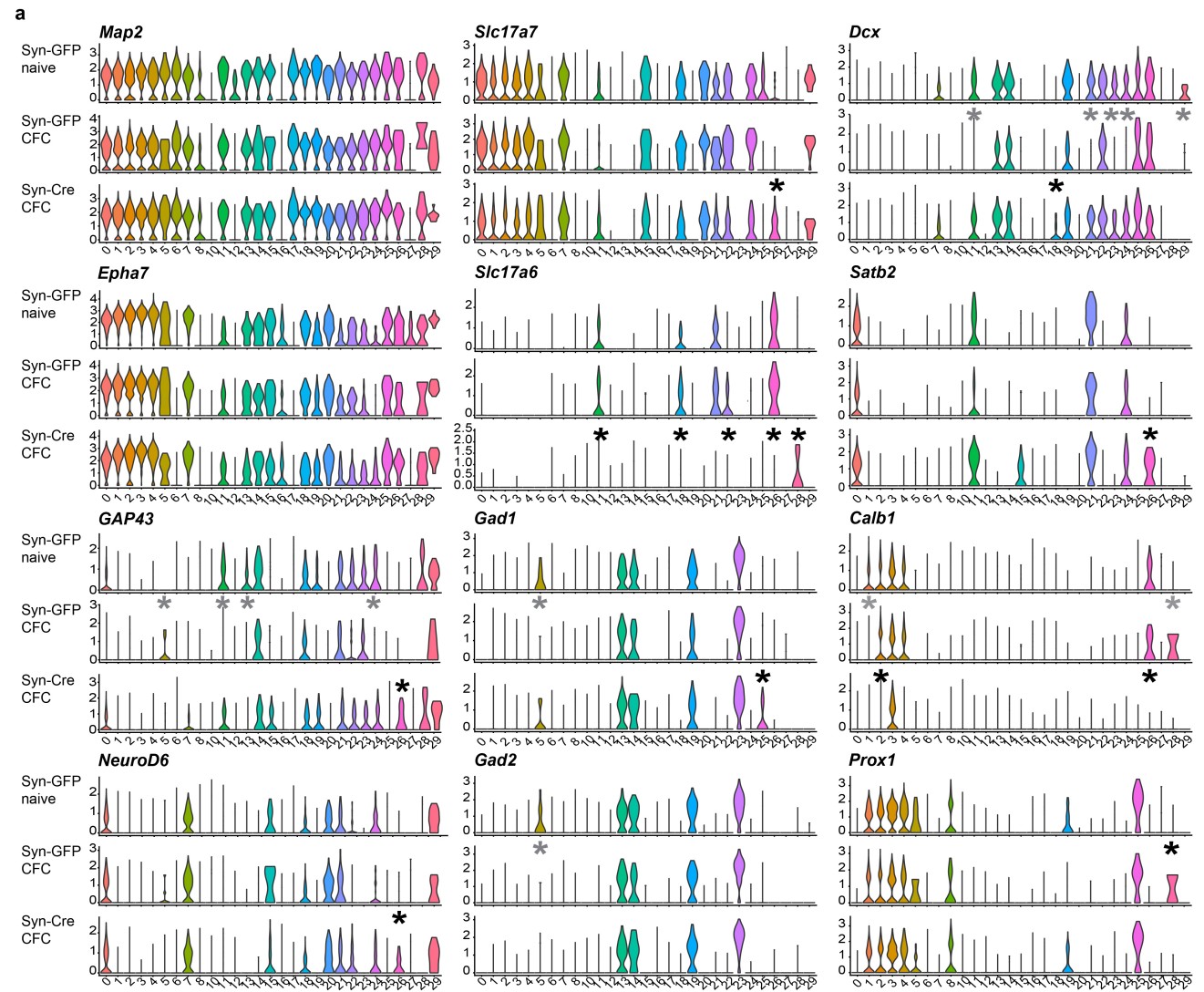

**Extended Data Fig. 8 | Cluster-specific changes of cellular phenotypes after CFC and *Tlr9* KO. a** CFC induced decrease of GAP43 from excitatory clusters 11 and 24, but the most dramatic effect was the disappearance of Dcx from most excitatory clusters (7, 11, 20, 24, 29) and inhibitory cluster 19 (gray stars). This CFC effect was prevented by *Tlr9* KO (black stars), enabling these clusters to retain Dcx while losing Slc17a6. In response to CFC, the most interesting change was in cluster 28, which shifted from an undifferentiated to a mature DGGC phenotype (Prox1+ Calb1+ Dcx−), an effect prevented by *Tlr9* KO. In response to Tlr9KO, significant phenotype changes were noted in immature DGGC (cluster 25), which lost their excitatory and acquired an inhibitory phenotype, and in excitatory cluster 26, which switched from an undifferentiated vGlut2+ Calb1+ GAP43− NeuroD6− phenotype to a vGlut1+ Satb2+ GAP43+ NeuroD6+ phenotype typical of CA neurons.

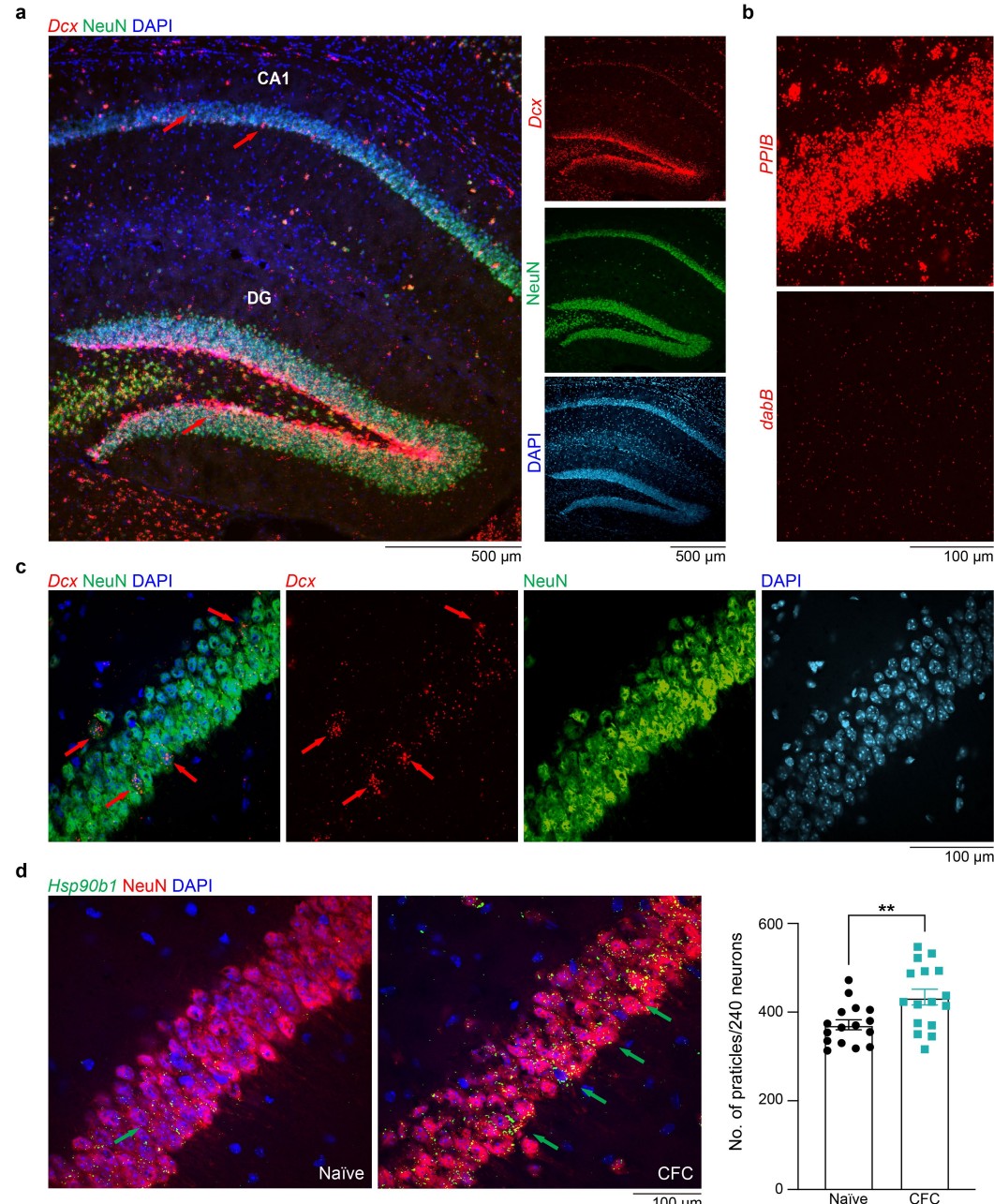

**Extended Data Fig. 9 | Validation of the RNA-seq data using RNAscope.**
**a** Low magnification images demonstrating Dcx mRNA expression in newly born DGGC as well as in individual CA1 neurons. **b** Positive and negative controls (see Methods for detailed description). **c** High magnification images identifying Dcx⁺ CA1 neurons. **d** The number of Hsp90b1 signals was significantly higher in hippocampi 96 h after CFC relative to the naïve group (left, representative micrographs; right, $n$ = 4 mice/group, 4 sections/mouse, 180-240 nuclei/mouse, two-tailed unpaired $t$ test, $t_{30}$ = 2.939, p = 0.0063). Data represented as mean ± s.e.m., $^{ns}$p > 0.05; ****p < 0.0001; ***p < 0.001; **p < 0.01; *p < 0.05.

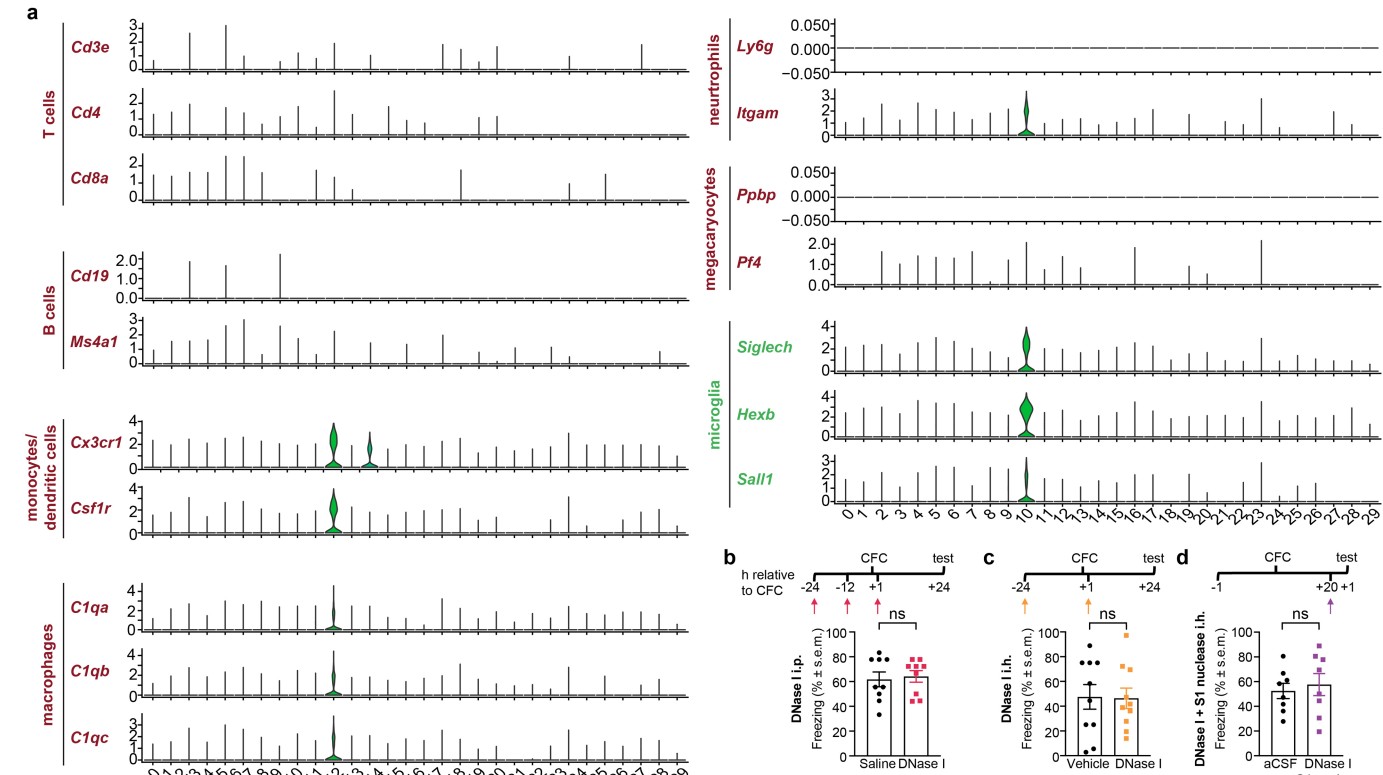

**Extended Data Fig. 10 | Analysis of the potential contribution of infiltrating immune cells and cell free DNA to the cellular and behavioral effects of CFC and Tlr9. a** Violin plots demonstrating lack of detectable infiltrating lymphocytes and myeloid cells in the hippocampus after CFC. Cells positive for macrophage/monocyte markers were identified as microglia based on the presence of Siglech, Hexb, and Sall1. **b** Intraperitoneal (i.p.) injections of DNase I before CFC did not affect freezing behavior at test ($n$ = 9 mice/group; two-tailed unpaired $t$ test, $t_{16}$ = 0.3194, p = 0.7536). **c** Similar freezing was also found in mice injected with intrahippocampal (i.h.) DNase I or vehicle before CFC ($n$ = 7 mice/group; two-tailed unpaired $t$ test, $t_{12}$ = 0.9197, p = 0.3758). **d** Combined i.h. injections of DNase I and S1 nuclease before the memory test did not affect context memory retrieval relative to vehicle controls ($n$ = 8 mice/group; two-tailed unpaired $t$ test, $t_{14}$ = 0.4799, p = 0.6387). Data represented as mean ± s.e.m., $^{ns}$p > 0.05; ****p < 0.0001; ***p < 0.001; **p < 0.01; *p < 0.05.

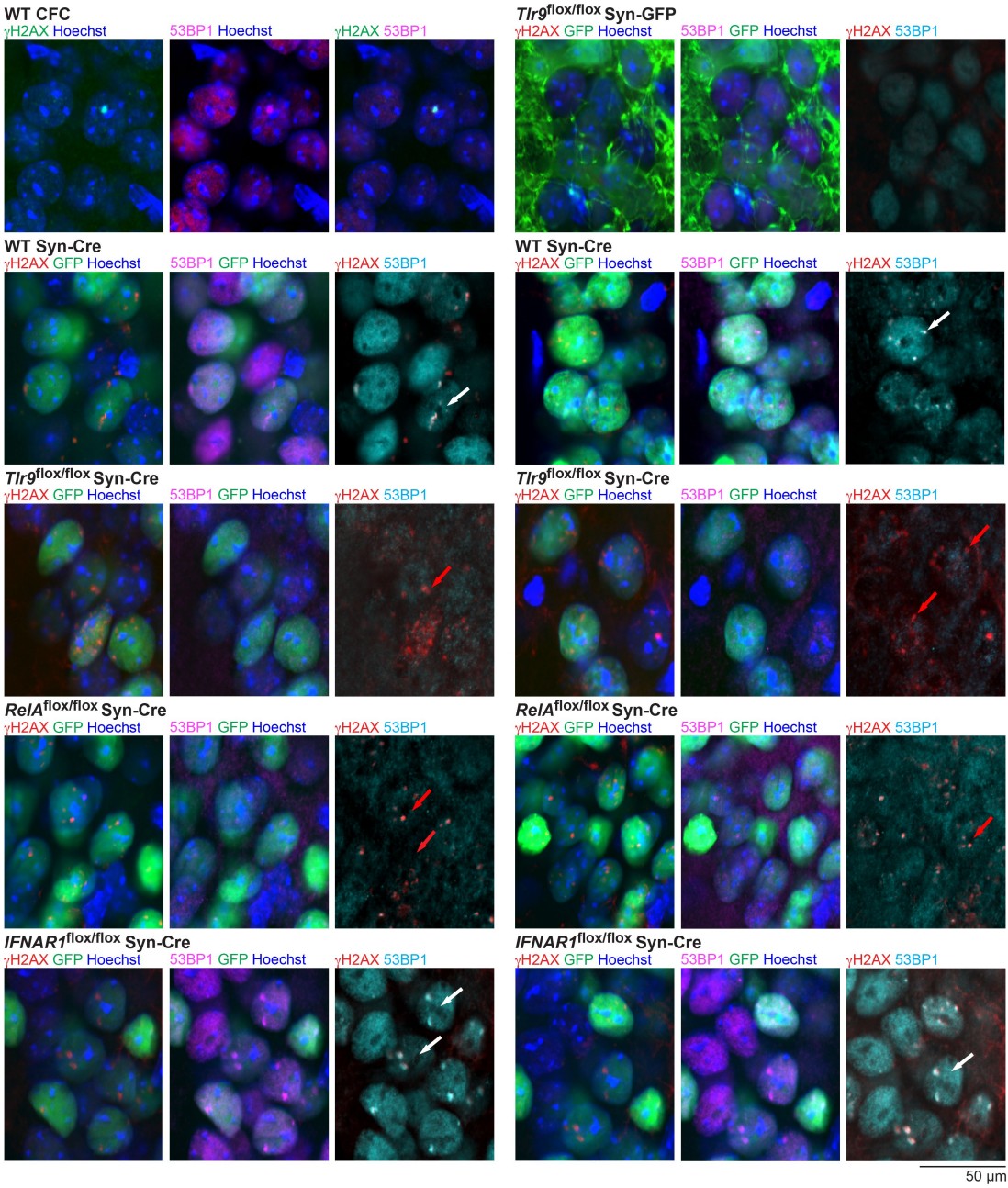

**Extended Data Fig. 11 | Low magnification images demonstrating the extent of genotype-dependent effects of Syn-Cre on dsDNA breaks and 53BP1 recruitment.** Images obtained from two mice per genotype depicting dsDNA breaks (γH2AX) and DDR in neuron-specific knockout of *Tlr9* and its downstream signaling genes *RelA* and *IFNAR1*. Red arrows indicate γH2AX signals and white arrows indicate γH2AX/53BP1 co-localization. Note lack of colocalization in hippocampi of Tlr9^flox/flox and RelA^flox/flox mice injected with Syn-Cre.

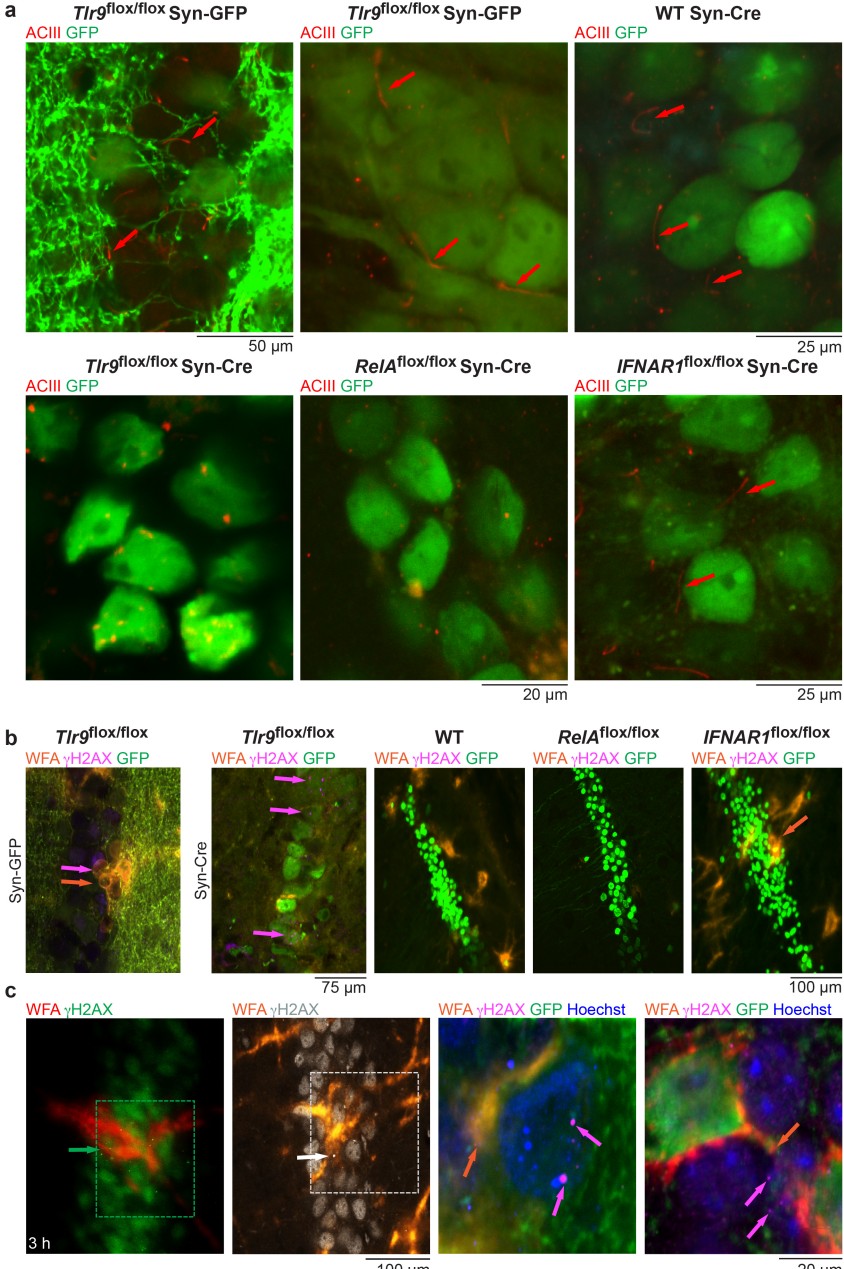

**Extended Data Fig. 12 | Low magnification images demonstrating the extent of genotype-dependent effects of Syn-Cre on ciliogenesis and PNN formation. a** Genotype-dependent effects of Syn-Cre on ciliogenesis showing lack of cilia (ACIII labeled filaments, red arrows) in Tlr9 KO and RelA KO hippocampi. **b** Genotype-dependent effects of Syn-Cre on PNN formation (orange arrows depict individual PNNs, magenta arrows indicate dsDNA breaks) revealing disappearance of PNNs in Tlr9 KO and RelA KO hippocampi. **c** PNN formation around CA1 clusters of intact WT mice exhibiting dsDNA breaks during memory consolidation (3 h after CFC) indicated with arrows depicting γH2AX and WFA signals according to the immunolabeling color code.

# Reporting Summary

## Statistics

For all statistical analyses, confirm that the following items are present in the figure legend, table legend, main text, or Methods section.

| n/a | Confirmed | |
|---|---|---|
| ☐ | ☒ | The exact sample size (*n*) for each experimental group/condition, given as a discrete number and unit of measurement |
| ☐ | ☒ | A statement on whether measurements were taken from distinct samples or whether the same sample was measured repeatedly |
| ☐ | ☒ | The statistical test(s) used AND whether they are one- or two-sided<br>*Only common tests should be described solely by name; describe more complex techniques in the Methods section.* |
| ☐ | ☒ | A description of all covariates tested |
| ☒ | ☐ | A description of any assumptions or corrections, such as tests of normality and adjustment for multiple comparisons |
| ☐ | ☒ | A full description of the statistical parameters including central tendency (e.g. means) or other basic estimates (e.g. regression coefficient) AND variation (e.g. standard deviation) or associated estimates of uncertainty (e.g. confidence intervals) |
| ☐ | ☒ | For null hypothesis testing, the test statistic (e.g. *F*, *t*, *r*) with confidence intervals, effect sizes, degrees of freedom and *P* value noted<br>*Give P values as exact values whenever suitable.* |
| ☒ | ☐ | For Bayesian analysis, information on the choice of priors and Markov chain Monte Carlo settings |
| ☒ | ☐ | For hierarchical and complex designs, identification of the appropriate level for tests and full reporting of outcomes |
| ☐ | ☒ | Estimates of effect sizes (e.g. Cohen's *d*, Pearson's *r*), indicating how they were calculated |

*Our web collection on statistics for biologists contains articles on many of the points above.*

## Software and code

Policy information about availability of computer code

| | |
|---|---|
| Data collection | snRNA-seq libraries were generated from about 8,000 individual cells captured in an oil emulsion on a Chromium Controller (10x Genomics, Pleasanton, CA.) cDNA was generated in the individual cell-gel bead emulsion micro-reactors while adding barcodes at the cellular and molecular level using the Chromium Next GEM Single Cell 3' Kit v3.1(10x Genomics kit 1000268.) The barcoded cDNAs from the individual cells were combined for the remaining library process. The unique molecular barcodes (UMIs) ensure that amplification artifacts will not skew the analysis. The libraries were analyzed on a Fragment Analyzer 5200 (Agilent Technologies, Santa Clara, CA) to ensure a normal size distribution with an average size of 450 bp. The libraries were sequenced on an llumina Sequencer with the following read lengths: 28 bp for Read 1, 10bp for i7 Index, 10bp for i5 Index and 90 bp for Read 2 at a read depth of 20,000 reads per cell. |
| Data analysis | Statistical analysis:<br>GraphPad Prism Version 10.1.0<br>Power analysis calculator:  http://www.stat.ubc.ca/~rollin/stats/ssize/n2.html<br><br>Sequencing Data Analysis:<br>SAMtools flagstat<br><br>Gene Expression Analysis:<br>STRING database (version 11.0)<br>GeneMANIA prediction serve<br>Mouse Genome Database (MGI) Batch Query Genome Analysis Tool (v6.23)<br>PANTHER Classification System engine, version 17.0<br>reactome.org, Version 86 |

R:

MEDIPS R package
Cellranger software v7.0.1
Seurat R package v4.3.0
scDblFinder v1.13.13
SoupX v1.6.2
Seurat SCTransform function
fgsea v1.20.0
Images:
Fiji Image J (NIH, vl.53f51, RRID:SCR_003070)

Figures:
Adobe Illustrator CS6 (Adobe, v27.5, RRID: SCR_010279)
Biorender.com

Code: GitHub (https://github.com/RadulovicLab/Nature-2024)

For manuscripts utilizing custom algorithms or software that are central to the research but not yet described in published literature, software must be made available to editors and reviewers. We strongly encourage code deposition in a community repository (e.g. GitHub). See the Nature Portfolio guidelines for submitting code & software for further information.

## Data

Policy information about availability of data

All manuscripts must include a data availability statement. This statement should provide the following information, where applicable:
- Accession codes, unique identifiers, or web links for publicly available datasets
- A description of any restrictions on data availability
- For clinical datasets or third party data, please ensure that the statement adheres to our policy

The bulk RNA-seq data are available at the NCBI Gene Expression Omnibus database under accession numbers GEO: GSE174076. The snRNA-seq data are available at the NCBI Gene Expression Omnibus database under accession numbers GEO: GSE254780. Source Data are provided with this paper. The mouse genome mus musculus mm10 was referenced for snRNA-seq. shRNA sequences were provided at GitHub (https://github.com/RadulovicLab/Nature-2024)

## Human research participants

Policy information about studies involving human research participants and Sex and Gender in Research.

| | |
|---|---|
| Reporting on sex and gender | N/A |
| Population characteristics | N/A |
| Recruitment | N/A |
| Ethics oversight | N/A |

Note that full information on the approval of the study protocol must also be provided in the manuscript.

# Field-specific reporting

Please select the one below that is the best fit for your research. If you are not sure, read the appropriate sections before making your selection.

☒ Life sciences    ☐ Behavioural & social sciences    ☐ Ecological, evolutionary & environmental sciences

For a reference copy of the document with all sections, see nature.com/documents/nr-reporting-summary-flat.pdf

# Life sciences study design

All studies must disclose on these points even when the disclosure is negative.

| | |
|---|---|
| Sample size | Statistical power to detect anticipated effect sizes was determined using power analysis (calculator at http://www.stat.ubc.ca/~rollin/stats/ssize/n2.html) conducted on representative samples of previous work and pilot experiments. For all proposed experiments, minimum power is set at 0.90 to detect an alpha = 0.05 (2 - sided test) for a difference in means from 20% to 40%, with a 15% common standard deviation. |

| | |
|---|---|
| Data exclusions | Mice with misplaced virus infusion or cannulas, and mice with less than 70% viral expression in the CAI were excluded. These exclusions are pre–established within the field and previously published by our lab and others. Cells with less than 1000 detected genes, more than 4000 detected genes, or larger than 5 percent mitochondrial gene percentage were excluded for further analysis, as according to established exclusion criteria for snRNA–seq. |
| Replication | All significant behavioral effects were replicated at least 3 times using WT, virus specific, and cell specific control groups. All cellular and molecular effects were shown in 6-12 biological replicates. All main gene expression effects were replicated with four different approaches (bulk RNA-seq, qPCR arrays, qPCR, and snRNA-seq). Replicates produced similar results relative to initial/representative experiments. |
| Randomization | With all genetic lines bred in our facility, littermates were randomly assigned to different experimental groups to minimize litter effects. |
| Blinding | The behavioral tests were performed blindly, either by experimentalists who were unaware of the treatments because the solutions were coded or by experimenters unaware of the experimental design. The experimenter performing the tests was not aware of the numbering code. Immunohistochemical analyses were performed by automated software (ImageJ). |

# Reporting for specific materials, systems and methods

We require information from authors about some types of materials, experimental systems and methods used in many studies. Here, indicate whether each material, system or method listed is relevant to your study. If you are not sure if a list item applies to your research, read the appropriate section before selecting a response.

## Materials & experimental systems

| n/a | Involved in the study |
|---|---|
| ☐ | ☒ Antibodies |
| ☐ | ☒ Eukaryotic cell lines |
| ☒ | ☐ Palaeontology and archaeology |
| ☐ | ☒ Animals and other organisms |
| ☒ | ☐ Clinical data |
| ☒ | ☐ Dual use research of concern |

## Methods

| n/a | Involved in the study |
|---|---|
| ☒ | ☐ ChIP-seq |
| ☐ | ☒ Flow cytometry |
| ☒ | ☐ MRI-based neuroimaging |

## Antibodies

| | |
|---|---|
| Antibodies used | Please see **Supplementary Methods** summarizing all relevant details and specifying recombinant, knock-out validated, or enhanced validation. For all other antibodies, we used pre-absorbtion or secondary only antibody controls. For the DNase antibody, we used DNase treatment as a control, for Tlr9 and RelA we used knockout tissue as controls. |
| Validation | Please see **Supplementary Methods** for a full list of primary antibodies with validation methods and/or relevant citations. |

## Eukaryotic cell lines

Policy information about cell lines and Sex and Gender in Research

| | |
|---|---|
| Cell line source(s) | *State the source of each cell line used and the sex of all primary cell lines and cells derived from human participants or vertebrate models.* |
| Authentication | *Describe the authentication procedures for each cell line used OR declare that none of the cell lines used were authenticated.* |
| Mycoplasma contamination | *Confirm that all cell lines tested negative for mycoplasma contamination OR describe the results of the testing for mycoplasma contamination OR declare that the cell lines were not tested for mycoplasma contamination.* |
| Commonly misidentified lines (See ICLAC register) | *Name any commonly misidentified cell lines used in the study and provide a rationale for their use.* |

## Palaeontology and Archaeology

| | |
|---|---|
| Specimen provenance | *Provide provenance information for specimens and describe permits that were obtained for the work (including the name of the issuing authority, the date of issue, and any identifying information). Permits should encompass collection and, where applicable, export.* |
| Specimen deposition | *Indicate where the specimens have been deposited to permit free access by other researchers.* |

| Dating methods | *If new dates are provided, describe how they were obtained (e.g. collection, storage, sample pretreatment and measurement), where they were obtained (i.e. lab name), the calibration program and the protocol for quality assurance OR state that no new dates are provided.* |

☐ Tick this box to confirm that the raw and calibrated dates are available in the paper or in Supplementary Information.

| Ethics oversight | *Identify the organization(s) that approved or provided guidance on the study protocol, OR state that no ethical approval or guidance was required and explain why not.* |

Note that full information on the approval of the study protocol must also be provided in the manuscript.

# Animals and other research organisms

Policy information about studies involving animals; ARRIVE guidelines recommended for reporting animal research, and Sex and Gender in Research

| Laboratory animals | Eight-week-old male and female C57BL/6N mice were obtained from Envigo. Tlr9$^{flox/flox}$ (C57BL/6J-Tlr9em1Ldm/J) mice with $loxP$ sites flanking exon 1 of theTlr9, IFNAR$^{flox/flox}$ (B6(Cg)-Ifnar1tm1.1Ees/J) mice with $loxP$ sites flanking exon 3 of the type-1 interferon α/β receptor gene, RelA$^{flox/flox}$ (B6.129S1-Relatm1Ukl/J) mice with $loxP$ sites flanking exon 1 of the Rela gene, and STING knockout mice C57BL/6J-Sting1gt/J; Goldenticket (Tmem173gt) were obtained from Jackson Laboratory and bred in institutional facilities. All mice were 8 weeks of age at the beginning of experiments. All mice were group housed (12 h light/dark cycle with lights on at 7 AM, temperature 20-22°C, humidity 30-60%) with ad libitum access to food and water (mice were switched to single housing one-week prior to experimentation). |
| Wild animals | No wild animals were used in this study. |
| Reporting on sex | All experimental groups were mixed sex, consisting of ~equal number of males and females. |
| Field-collected samples | No field collected samples were used in this study. |
| Ethics oversight | All procedures were approved by Northwestern University's Animal Care and Use Committee and Albert Einstein's Animal Care and Use Committee in compliance with US National Institutes of Health standards, and Aarhus University in compliance with the Danish National Animal Experiment Committee. |

Note that full information on the approval of the study protocol must also be provided in the manuscript.

# Clinical data

Policy information about clinical studies
All manuscripts should comply with the ICMJE guidelines for publication of clinical research and a completed CONSORT checklist must be included with all submissions.

| Clinical trial registration | *Provide the trial registration number from ClinicalTrials.gov or an equivalent agency.* |
| Study protocol | *Note where the full trial protocol can be accessed OR if not available, explain why.* |
| Data collection | *Describe the settings and locales of data collection, noting the time periods of recruitment and data collection.* |
| Outcomes | *Describe how you pre-defined primary and secondary outcome measures and how you assessed these measures.* |

# Dual use research of concern

Policy information about dual use research of concern

## Hazards

Could the accidental, deliberate or reckless misuse of agents or technologies generated in the work, or the application of information presented in the manuscript, pose a threat to:

| No | Yes | |
|----|-----|---|
| ☒ | ☐ | Public health |
| ☒ | ☐ | National security |
| ☒ | ☐ | Crops and/or livestock |
| ☒ | ☐ | Ecosystems |
| ☒ | ☐ | Any other significant area |

## Experiments of concern

Does the work involve any of these experiments of concern:

No  Yes

☒ ☐ Demonstrate how to render a vaccine ineffective
☒ ☐ Confer resistance to therapeutically useful antibiotics or antiviral agents
☒ ☐ Enhance the virulence of a pathogen or render a nonpathogen virulent
☒ ☐ Increase transmissibility of a pathogen
☒ ☐ Alter the host range of a pathogen
☒ ☐ Enable evasion of diagnostic/detection modalities
☒ ☐ Enable the weaponization of a biological agent or toxin
☒ ☐ Any other potentially harmful combination of experiments and agents

# ChIP-seq

## Data deposition

☐ Confirm that both raw and final processed data have been deposited in a public database such as GEO.

☐ Confirm that you have deposited or provided access to graph files (e.g. BED files) for the called peaks.

Data access links
May remain private before publication.

*For "Initial submission" or "Revised version" documents, provide reviewer access links. For your "Final submission" document, provide a link to the deposited data.*

Files in database submission

*Provide a list of all files available in the database submission.*

Genome browser session
(e.g. UCSC)

*Provide a link to an anonymized genome browser session for "Initial submission" and "Revised version" documents only, to enable peer review. Write "no longer applicable" for "Final submission" documents.*

## Methodology

Replicates

*Describe the experimental replicates, specifying number, type and replicate agreement.*

Sequencing depth

*Describe the sequencing depth for each experiment, providing the total number of reads, uniquely mapped reads, length of reads and whether they were paired- or single-end.*

Antibodies

*Describe the antibodies used for the ChIP-seq experiments; as applicable, provide supplier name, catalog number, clone name, and lot number.*

Peak calling parameters

*Specify the command line program and parameters used for read mapping and peak calling, including the ChIP, control and index files used.*

Data quality

*Describe the methods used to ensure data quality in full detail, including how many peaks are at FDR 5% and above 5-fold enrichment.*

Software

*Describe the software used to collect and analyze the ChIP-seq data. For custom code that has been deposited into a community repository, provide accession details.*

# Flow Cytometry

## Plots

Confirm that:

☒ The axis labels state the marker and fluorochrome used (e.g. CD4-FITC).

☒ The axis scales are clearly visible. Include numbers along axes only for bottom left plot of group (a 'group' is an analysis of identical markers).

☒ All plots are contour plots with outliers or pseudocolor plots.

☒ A numerical value for number of cells or percentage (with statistics) is provided.

## Methodology

Sample preparation

FANS was only used for cell-sorting and not for immuno-fluorescence analysis (samples were used for sn-RNA-Seq).

Instrument

Beckman Coulter MoFloXDP cell sorter

| Software | FlowJo v10.9 |

| Cell population abundance | Nuclei were counted with a hemocytometer followed by flow cytometry (FCM) during sorting (source data for ED Fig.7). |

| Gating strategy | The sample was gated on the nuclei population based on the forward and side scatter densityFSC/SSC distribution, gated on SSC-W/SSC-H to get rid of the aggregates, gated on DAPI+ nuclei, further getting grid of aggregates, overlaid GFP+ with ungated population to verify the accuracy of nuclei gating. |

☒ Tick this box to confirm that a figure exemplifying the gating strategy is provided in the Supplementary Information.

# Magnetic resonance imaging

## Experimental design

| Design type | *Indicate task or resting state; event-related or block design.* |

| Design specifications | *Specify the number of blocks, trials or experimental units per session and/or subject, and specify the length of each trial or block (if trials are blocked) and interval between trials.* |

| Behavioral performance measures | *State number and/or type of variables recorded (e.g. correct button press, response time) and what statistics were used to establish that the subjects were performing the task as expected (e.g. mean, range, and/or standard deviation across subjects).* |

## Acquisition

| Imaging type(s) | *Specify: functional, structural, diffusion, perfusion.* |

| Field strength | *Specify in Tesla* |

| Sequence & imaging parameters | *Specify the pulse sequence type (gradient echo, spin echo, etc.), imaging type (EPI, spiral, etc.), field of view, matrix size, slice thickness, orientation and TE/TR/flip angle.* |

| Area of acquisition | *State whether a whole brain scan was used OR define the area of acquisition, describing how the region was determined.* |

Diffusion MRI     ☐ Used     ☐ Not used

## Preprocessing

| Preprocessing software | *Provide detail on software version and revision number and on specific parameters (model/functions, brain extraction, segmentation, smoothing kernel size, etc.).* |

| Normalization | *If data were normalized/standardized, describe the approach(es): specify linear or non-linear and define image types used for transformation OR indicate that data were not normalized and explain rationale for lack of normalization.* |

| Normalization template | *Describe the template used for normalization/transformation, specifying subject space or group standardized space (e.g. original Talairach, MNI305, ICBM152) OR indicate that the data were not normalized.* |

| Noise and artifact removal | *Describe your procedure(s) for artifact and structured noise removal, specifying motion parameters, tissue signals and physiological signals (heart rate, respiration).* |

| Volume censoring | *Define your software and/or method and criteria for volume censoring, and state the extent of such censoring.* |

## Statistical modeling & inference

| Model type and settings | *Specify type (mass univariate, multivariate, RSA, predictive, etc.) and describe essential details of the model at the first and second levels (e.g. fixed, random or mixed effects; drift or auto-correlation).* |

| Effect(s) tested | *Define precise effect in terms of the task or stimulus conditions instead of psychological concepts and indicate whether ANOVA or factorial designs were used.* |

Specify type of analysis:     ☐ Whole brain     ☐ ROI-based     ☐ Both

| Statistic type for inference (See Eklund et al. 2016) | *Specify voxel-wise or cluster-wise and report all relevant parameters for cluster-wise methods.* |

| Correction | *Describe the type of correction and how it is obtained for multiple comparisons (e.g. FWE, FDR, permutation or Monte Carlo).* |

## Models & analysis

| n/a | Involved in the study |
|-----|------------------------|
| ☐ | ☐ Functional and/or effective connectivity |
| ☐ | ☐ Graph analysis |
| ☐ | ☐ Multivariate modeling or predictive analysis |

**Functional and/or effective connectivity**

*Report the measures of dependence used and the model details (e.g. Pearson correlation, partial correlation, mutual information).*

**Graph analysis**

*Report the dependent variable and connectivity measure, specifying weighted graph or binarized graph, subject- or group-level, and the global and/or node summaries used (e.g. clustering coefficient, efficiency, etc.).*

**Multivariate modeling and predictive analysis**

*Specify independent variables, features extraction and dimension reduction, model, training and evaluation metrics.*

