## [Peer Review File · Nature]

Manuscript Title: Formation of Memory Assemblies through the DNA Sensing Tlr9 Pathway

Redactions – unpublished data

Reviewer Comments & Author Rebuttals

Reviewer Reports on the Initial Version:

Referees' comments:

Referee #1 (Remarks to the Author):

Summary:

This manuscript investigates the role of Tlr9 signaling in fear memory formation. The experiments in this manuscript examine the role of Tlr9 and downstream inflammatory mediators and DNA repair mechanisms in memory formation, a mechanistic pathway that is largely unexplored in the brain. The authors seek to demonstrate that these mechanisms occur within a discrete subset of neurons that represent the fear memory, as the mechanisms by which specific populations of neurons are committed to represent a single memory remain largely unknown. However, the authors do not provide convincing evidence that mechanisms downstream of Tlr9 signaling perform this function. Also, although this manuscript has the potential to provide some insight into the mechanistic role of Tlr9-mediated inflammatory signaling in memory formation, it is neither technically nor conceptually novel.

Major Concerns:

1. The data presented does not support the claim that sustained Tlr9 signaling and downstream activity marks a neuronal assembly committed to the formation of a single memory. Tlr9 upregulation is observed following CFC in bulk hippocampal sequencing, and CFC behavior is performed following virally mediated Tlr9 knockout in all CA1 neurons. Furthermore, γ H2AX, a downstream effector of Tlr9 signaling, was not selectively present within cFos-expressing neurons: 52% of cFos positive nuclei were γ H2AX negative.
2. The time following fear conditioning that molecular analyses were performed was inconsistent throughout the manuscript, which raises doubt about the rationale for individual experiments and the conclusions that can be drawn by considering the entire dataset. For example, RNA sequencing in Figure 1a compares gene expression at 4 days and 28 days following CFC (it is also unclear if this differential expression data is relative to a home cage control). The 28-day remote timepoint was not considered in the remainder of the manuscript. The peak of γ H2AX expression was then observed to be 1 hour following CFC, and molecular analyses in Figure 3 were then performed at this timepoint to detect overlap with immediate early genes. The timepoints for molecular analyses in Figures 4 and 5 were not clearly stated.
3. The experimental timeline in Figure 4 is not stated. Necessary details for this experiment include the age at which surgery was performed, the time allotted for recovery, and the timepoint at which tissue was taken following CFC.
4. Number of mice is not listed for experiments in Figure 4. Thus, it appears that the data shown is derived from one biological replicate.
5. CFC behavior data in Figure 5 should include control experiments for both cued fear conditioning and short-term memory.

6. The neuronal specificity of the effect of Tlr9 deletion on CFC is supported using an astrocyte-specific Cre driver as a control (Figure 5b). Given the inflammatory mechanisms downstream of Tlr9, the authors must include a microglia-specific Cre driver as an additional control.

7. Figure 1 (panels a, b, and d) is not legible in the resolution provided, which precludes full review of this manuscript.

Minor Concerns:

1. The analysis of Figure 3d appears to be at the level of individual neurons (this is unclear), although biological replicates for this experiment should be at the level of mice (n=6).

2. DNase2A shRNA in Figure 5d is driven by the U6 promoter, which would not result in a neuron-specific knockdown.

3. AAV vectors for Syn-GFP and Syn-Cre are listed as serotype 9 in the text but serotype 8 in Figure 5a.

Referee #2 (Remarks to the Author):

This study provides interesting observations on the nature of long term memory formation via innate immune sensing in the cytosol of DNA double-strand breaks. DNA damage in neurons was essential for memory formation to a fear stimulus and required a purported Tlr9-mediated DNA sensing and repair machinery. TLR9 was also required for ciliogenesis and PNN build-up in proximity to neurons with dsDNA breaks. Stable memory representation was associated with TLR9 involvement but not the other cytosolic DNA sensor cGAS/STING, implicating DNA recognition in endolysosomes. Interestingly, TLR9 deficiency resulted in persistent breaks and failure to establish memory. The authors conclude that TLR9 directs a DNA repair response that is necessary for memory formation. Consistent with this model, the authors show that DNase2 knockdown in neurons also prevented memory whereas TREX1 overexpression did not. In addition, a TLR9 antagonist prevented memory and a cGAS-STING inhibitor did not.

There is considerable novelty to the model presented in this study and potentially broad interest if the findings are correct. However, I have concerns regarding the experimental design for several key figures. Some of the data also raises issues regarding accuracy. These concerns must be addressed prior to further consideration.

1. [REDACTED]

2. [REDACTED]

3. Fig 4s and 5. AAV9 syn-cre is used to remove TLR9 but is not used on control animals. This delivery method results in high expression of cre, which is known to cause extensive DNA damage characterized by gH2AX foci and breaks found in metaphase chromosome spreads (Silver and Livingston Mol Cell 2001 PMID11511376). This issue of directing cre recombinase in the floxed TLR9 mice but not in controls severely confounds the results. The appropriate control is AAV9 syn-cre in mice that do not contain floxed TLR9. This was not performed. For example, Fig 5h shows many more gH2AX foci in the syn-cre treated TLR9 floxed mice, which may simply reflect off target activity of syn-cre. Again, the more appropriate control would be to treat mice lacking a floxed allele with Syn-cre. These issues pose serious concerns about data interpretation.

4. [REDACTED]

5. No independent methods are used to demonstrate that TLR9 actually directs DNA repair in neurons.

6. A critical question is how TLR9 specifically recognizes cytosolic DNA and cGAS does not. Is there an endosomal context of dsDNA recognition that is responsible for this specificity? A simple staining experiment by immunofluorescence could be used to address this point.

Referee #3 (Remarks to the Author):

Jovasevic and colleagues suggest that neuronally expressed TLR9 controls a DNA repair response critical for memory formation and perineuronal nets in interneurons and regulates ciliogenesis. Cell-autonomous TLR9-mediated sensing of DNA damage in neurons that counterbalances damage-induced neurocognitive deficits would be a new concept of broad relevance. While this novel concept would be intriguing and of interest to a more general readership, this reviewer is not convinced that the data support these broad claims.

There are several significant issues that need to be addressed:

1. The authors perform bulk RNA sequencing of mouse hippocampi after Contextual Fear Conditioning and find differences in gene expression profiles after four days signifying an immune response. The data's bioinformatics analysis demonstrates the upregulation of TLR pathway genes and 'most prominently' CD14. It is well established that stress and CRC activate microglia cells, which trigger the migration of monocytes into the brain. Hence, it is likely that the observed upregulation of innate immune-related genes represents increased recruitment of monocytes to the brain rather than upregulation of TLR9 in neurons. The authors need to demonstrate that CRC increases TLR9 and pathway components in neurons using a different approach, such as single-cell sequencing or spatial biologic methods.
2. TLR9 is an endosomal protein that senses DNA in the endosomal compartment. Thus, there is a conceptual problem that needs to be addressed. How does TLR9 sense DNA damage in neurons in a cell-autonomous fashion, as proposed by the authors?
3. What is the evidence that TLR9 senses DNA damage in neurons? No direct data show that TLR9 activation occurs upon DNA damage. Only correlations are shown.
4. It is well established that AAVs are potent activators of type I production via mainly a TLR9-dependent mechanism. The in vivo work to reduce TLR9 expression in neurons is performed using AAV viral transduction of cells. Since AAVs can readily activate the TLR9-MyD88 pathway, this approach is highly prone to artifacts. The industry struggles with this issue for the translation of AAV-mediated gene delivery. Even the CpG content difference in the delivered genes versus controls might make a significant difference. Hence, this reviewer remains unconvinced by the neuron-specific deletion approach taken by the authors, as bystander cell TLR9 activation by the AAV vector could influence the phenotype.

Minor issues:

1. The bioinformatics approach to data analysis remains obscure. Please describe the bioinformatic pipeline in detail. Fig. 1 a,b, and d are not legible and do not contain useful information. For example, for any TLR or other gene, similar hubs of genes could be generated, as shown in 1b. What is the relevance of showing this other than convincing the reader that this is the essential gene? On what basis has TLR9 been selected for highlighting?
2. Imaging data require quantification.

Author Rebuttals to Initial Comments:

We thank the reviewers for their very helpful comments, suggestions, as well as the raised concerns. We hope that our extensive revisions, both in terms of added experimental data and text edits, adequately respond to them. Given that we add a large body of new analyses, we reorganized the manuscript, and removed parts of the earlier manuscript that did not directly add to the main points or that were too speculative.

A. General Revisions

1. Effects of Cre expression on dsDNA breaks

This is a major issue because most of our manipulations involved floxed genetic mouse lines and injection of Cre-expressing adeno-associated viruses. Given that Cre alone can induce considerable genomic instability in microglia, but also likely in neurons [Pollina et al., Nature 2023 (although the extent of Cre-induced dsDNA breaks relative to other manipulations was not directly compared)], we followed the advice of Reviewer 2 and now include data with WT mice injected with Syn-Cre.

1a. First, we would like to point out that we have carefully titrated the virus and optimized the infusion conditions before choosing the optimal dilution resulting in gene knockdown but not damaging hippocampal cells (1×10^{12} Vg/ml, 0.5 μ l over 10 min/site). This is an important point because titers of 5×10^{12} or more, or shorter infusion times resulted in neuron damage and at 1×10^{13} in neuron death.

1b. We show that indeed, injection of Cre increased the number of neurons with dsDNA breaks relative to Syn-GFP (**Fig. 5a**, Extended Data Fig. 13). Nevertheless, this effect was significantly smaller than the effects of various inflammatory gene knockouts. Also, our main findings demonstrate a selective disruption of the centrosomal DNA damage response (DDR) by Tlr9 and RelA knockdown (KO), and this response was unaffected in WT and IFNAR KO mice. This and other key effects (ciliogenesis, PNN) were also not affected by Cre expression. This is in line with our conclusion that disrupted DDR, ciliogenesis, and PNN build up are the key processes affected by Tlr9KO leading to genomic instability and impaired memory. To support this interpretation, we previously included a chronic stress experiment showing a large number of CA1 neurons with dsDNA breaks with not only intact, but even enhanced context memory. We removed these data to retain focus, so this is a reminder for the reviewers. If essential, we can provide those data again. In summary, although dsDNA breaks are the most likely trigger of DDR, our data indicate that the induction of breaks by themselves is not enough to disrupt learning as long as neuronal centrosome function remains intact.

1c. We show that Syn-Cre injection in WT mice does not cause memory impairments when compared to Syn-GFP injection (Extended Data Fig. 8c).

1d. We confirm the memory impairing effect of Tlr9 knockdown using a different approach, neuron-specific injection of Tlr9 shRNA (**Fig. 3c**).

2. Cell specificity

Another important point was the neuronal specificity of the observed effects as well as the specificity of our manipulations.

2a. We now provide data demonstrating that learning-induced activation of Tlr9 (determined by colocalization with Lamp2, **Fig. 1c,d**) and dsDNA breaks only in neurons but not in microglia or astrocytes in CA1 (**Fig. 2a**). We used Tlr9 colocalization with vesicle markers as an established measure of Tlr9 activity in response to nucleic acid sensing (Sasai et al., *Science*, 2010; Combes et al., *Nat Commun*, 2017). Overall, we were surprised by the low frequency of Tlr9 detection in non-neuronal cells that, even if present, did not show colocalization with late vesicles typical of active Tlr9.

2b. The reviewers were rightfully concerned that microglia could contribute to the inflammatory and behavioral responses seen after CFC. Indeed, gross assessment of astroglial reactivity in virus-injected mice confirmed such reaction (Extended Data Fig. 8b). However, there was no difference between control and Cre viruses and this level of reactivity was not likely to contribute to the behavioral effects of the Tlr9 knockdown. This was shown by astrocytic Tlr9 knockdown (**Fig. 3e**) and by depleting microglia (**Fig. 3f**), which did not affect CFC. We chose the depletion diet over virus manipulations in this study because most available microglial AAV drivers suffer from lack of specificity or efficiency. Even with such a drastic approach, CFC was unaffected, which is consistent with findings by other groups (Elmore et al., *Neuron* 2014).

2c. Remaining issues on cell-specific gene expression were addressed using snRNA-seq (see below). Here it is noteworthy that even if virus infusions had induced enhanced astroglial reactivity, this did not prevent the up-regulation of inflammation-related genes by CFC. Thus, there was no ceiling effect with this respect.

3. Demonstrating up-regulation of Tlr9 gene expression with single cell transcriptomics

We did not perform snRNA-seq initially because the transcripts of immune genes in the brain is very low under physiological conditions. Detailed information on that is now available at the Allen Brain Map site:

<https://portal.brain-map.org/atlas-and-data/rnaseq/mouse-whole-cortex-and-hippocampus-10x>

In response to reviewers' concerns we decided to follow their advice, first, because we could better define the inflammatory phenotype of individual populations, and second, because we could examine the direct effects of Tlr9KO on neuronal and non-neuronal cell phenotypes.

We obtained several lines of evidence supporting a neuron-specific inflammatory phenotype and a neuron-specific role of Tlr9:

3a. CFC induced a set of conserved genes directly involved in Tlr9 function (Hsp90b1, critical for activation by inducing proper Tlr9 folding; ATP6voc, needed for vesicle acidification and Tlr9 activation; IL-6, key downstream target of Tlr9 signaling) as well as genes involved in endoplasmic reticulum function, vesicle trafficking, and immune

function, among others (**Fig. 4b**). This fully supported the data presented in Fig. 1 with bulk RNA-seq and co-labeling of Tlr9 with Lamp2. Most robust changes of gene expression were found in CA excitatory neurons, DGGC, and interneurons, whereas only a few robust changes were found in non-neuronal populations (**Fig. 4a-e**, Extended Data Fig. 11,12).

3b. Reactome pathway analysis identified Dcx- excitatory neurons as the population showing the strongest inflammatory phenotype, including not only Tlr signaling and cytokine release, but also DDR and cilium assembly (**Fig. 4f**).

3c. Neuron-specific CFC-induced gene expression was prevented by Tlr9KO. Importantly, the KO did not disrupt gene expression in naïve controls that were not exposed to CFC (Supplementary Table 3).

3d. Cell specificity of the infection was confirmed by delineating neuron-specific from non-neuronal specific gene expression profiles in FANS-sorted samples (GFP+ and GFP- after Syn-Cre injection, Extended Data Fig. 10a).

3e. We were particularly thrilled to confirm the presence of many Dcx+ neurons within CA and DGGC neuron clusters that were clearly separate from immature DGGC (Zhou et al., Nature, 2022) and then to show that (i) this phenotype disappears in neurons within small clusters of CA1 pyramidal cells during memory formation, and (ii) that this effect is also Tlr9-dependent. We discuss these effects in the context of neuronal differentiation/maturity states, but also in the context of Dcx's role in nucleo-centrosomal coupling, which further supports the proposed role of Tlr9 in integrating nuclear dsDNA breaks with centrosome DDR during memory formation.

3f. We also demonstrate interesting effects of Tlr9 KO independent of CFC, such as disappearance of the vGlut2 CA neuron cluster, as well as appearance of a novel cluster with a paradoxical response to Tlr9 KO (increased gene expression).

Although we do not know at this time whether each population of these discrete CA1 neurons contributes memory and how, reporting our findings will serve as a guide for many groups in the memory field but also in the field of neuronal and cell biology more generally.

4. Quantification

We now provide detailed quantification and statistical analyses for all presented data. Also, we repeated some of the immunostainings so that added replicates and new experiments are compared under the same labeling conditions. Quantification procedures are described in detail and involve various functions and plug-ins of Image J as indicated.

B. Specific Responses

Reviewer 1

Overall, Reviewer 1 seemed the least enthusiastic about our report for reasons highlighted in the paragraph below:

This manuscript investigates the role of Tlr9 signaling in fear memory formation. The experiments in this manuscript *examine the role of Tlr9 and downstream inflammatory mediators* and DNA repair mechanisms in memory formation, a mechanistic pathway that is largely unexplored in the brain. The authors seek to demonstrate that these mechanisms occur within a discrete subset of neurons that represent the fear memory, as the mechanisms by which specific populations of neurons are committed to represent a *single memory* remain largely unknown. However, the authors do not provide convincing evidence *that mechanisms downstream of Tlr9 signaling* perform this function. Also, although this manuscript has the potential to provide some insight into the mechanistic role of Tlr9-mediated inflammatory signaling in memory formation, it is *neither technically nor conceptually novel*.

(i) We now clarified that the mechanisms that we identified are not meant to represent a single memory but a novel memory mechanism used by a subset of memory-processing neurons that do not show IEG response. Individual neurons might encode bits of specific information that is used across memory representations, but our work does not directly address that question. We have re-written any text containing ambiguity regarding this issue.

This mechanism that we demonstrate relies on activation of the dsDNA sensing Tlr9 pathway and Tlr9-mediated DDR and in small patchy clusters of CA1 pyramidal neurons. We do not know how this population adds to the various features of memory representations conferred by other populations (Creb, Egr1, cFos, Npas4, etc), but the newly added experiments with RAM labeling (**Fig. 2f**) identified a stable population that is not prone to mount IEG responses and thus IEG-associated plasticity both during memory formation and reactivation. Such population is thus likely to contribute to the stability of memory, but this needs further examination and direct comparison with other neuronal populations.

(ii) Although we have not invented a new technology to investigate the neuron-specific roles of Tlr9 signaling in learning and memory (given that this is the first demonstration for a memory role of Tlr9 signaling), we have used cutting-edge technologies to establish a role of Tlr9 in memory formation at the level of individual cells. We applied extremely laborious analyses of individual neurons using high magnification confocal microscopy that enabled us to study *in situ* the subcellular aspects of dsDNA breaks, ciliogenesis, DNA repair, and Tlr9 signaling. This approach was uniquely suitable for low abundance transcripts that can easily be overlooked with snRNA seq. It also allowed us to perform thorough cell specific analyses at a cellular and subcellular resolution that Chip-Seq lacks. Given that we now add a set of snRNA-seq experiments, which significantly add to this aspect of our work, confirm, and expand our earlier findings, we hope that Reviewer 1 will see this issue in a more positive way.

(iii) We are surprised that Reviewer 1 did not appreciate the conceptual novelty of our work given that:

1. We show for the first time that a neuron-specific DNA-sensing inflammatory pathway contributes to memory formation (especially given the prevalent view that inflammatory signaling causes tissue damage and neurodegeneration).

2. We demonstrate both by cloning, sequencing, and cellular analysis that, after learning, clusters of neurons enter a state that supports Tlr9 signaling, a state which entails persistent formation of dsDNA breaks (contrary to the view that most dsDNA breaks repair after physiological neuronal activity), nuclear envelope rupture, and extranuclear release of DNA and histone co-localizing in Tlr9-rich endomembrane compartments.

3. We show that these neurons are a population that poorly overlaps with IEG-inducing neurons, which indicates a novel memory mechanism.

4. We show a key role of Tlr9 in centrosomal recruitment of DDR and memory formation and identify several downstream mechanisms including ciliogenesis and PNN formation. This is now supplemented with data identifying RelA and Dcx but not interferon pathways as downstream mediators of Tlr9 actions.

5. We proposed a novel model of memory based on molecular/genetic and cellular changes occurring within hours and days of learning (now revised to also highlight Tlr-mediated changes of maturation/differentiation state in addition to Tlr-coordinated DDR).

6. It is particularly important to emphasize that the specific involvement of Tlr9 but not STING in neuronal genomic stability and memory can have profound translational implications for delineating adaptive from maladaptive inflammatory responses and for preserving neuronal health while devising anti-inflammatory treatment strategies for various brain disorders.

We hope that the additions, revisions, and clarifications introduced in the new version will excite the interest of Reviewer 1.

Major Concerns:

1. *“The data presented does not support the claim that sustained Tlr9 signaling and downstream activity marks a neuronal assembly committed to the formation of a **single** memory.”*

It seems that the word “committed” came across as if to mean “single”. This was not our intention. Whereas information coded by individual neurons can be used flexibly for diverse representations, we do not know at this time whether individual neurons code one or multiple types of information integrated in a memory representation. We have carefully re-written the paper to clarify this important point.

2. *“Tlr9 knockdown was induced in all neurons and not only in neurons showing DNA damage”*

We completely agree with Reviewer 1. At this time, we do not have the tools for such manipulations, and it will probably take some time, as it did for the IEG-based

approaches, to achieve that. Nevertheless, even when we manipulate individual populations, such as IEG-expressing neurons, we have no proof that all IEG responsive neurons or only a subset will contribute to memory formation. With all of these constraints in mind, we find it reasonable to believe that neurons in which Tlr9 is induced would be the ones most affected by the knockdown. We now support this assumption with the snRNA-Seq data showing discrete effects of Tlr9 in CFC-activated neuronal populations but not in neurons of non-conditioned controls (Supplementary Table 3).

3. *“Furthermore, γ H2AX, a downstream effector of Tlr9 signaling, was not selectively present within cFos-expressing neurons: 52% of cFos positive nuclei were γ H2AX negative.”*

This is indeed the case (although γ H2AX is an upstream activator of Tlr9). We now discuss more thoroughly this point (Discussion, p.16, pg2).

Briefly, many IEGs (Egr1, Arc, NPAS4) and other transcription factors (NFkB, RelA) are expressed in nonoverlapping neurons and yet contribute to memory formation. The finding that γ H2AX neurons are better aligned with inflammatory rather than IEG responses highlights the stronger association of dsDNA breaks with inflammatory signaling rather than IEG. Our view is that some combination of all these events determines the likelihood of neurons to be recruited to a memory assembly. We, again, must have confounded our view with the too liberal use of the term “commitment”, for which we apologize.

2. *The time following fear conditioning that molecular analyses were performed was inconsistent throughout the manuscript, which raises doubt about the rationale for individual experiments and the conclusions that can be drawn by considering the entire dataset. For example, RNA sequencing in Figure 1a compares gene expression at 4 days and 28 days following CFC (it is also unclear if this differential expression data is relative to a home cage control). The 28-day remote timepoint was not considered in the remainder of the manuscript.*

Because we observed maximal gene expression changes 4 days after CFC we had no rationale to study the 28 day time point in this study (in fact we recently published the results relating to remote memory in Jovasevic et al., iScience, 2022). However, we needed a time course study to identify the events leading to Tlr9 up-regulation. These occurred at various times, occasionally creating confusion. We now segregated early (1h-3h) and late (6h-96h) events whenever possible and add a schematic (Extended Data Fig. 15) summarizing their time course with respect to one another as they precede Tlr9 activation.

“The peak of γ H2AX expression was then observed to be 1 hour following CFC, and molecular analyses in Figure 3 were then performed at this timepoint to detect overlap with immediate early genes. The timepoints for molecular analyses in Figures 4 and 5 were not clearly stated.”

The time points for all figures are now provided in the lower left corner. We also indicate that all molecular studies are performed in hippocampi obtained 24h after the last memory test.

3. The experimental timeline in Figure 4 is not stated. Necessary details for this experiment include the age at which surgery was performed, the time allotted for recovery, and the timepoint at which tissue was taken following CFC.

We apologize for this omission. We now provide schematics for each manipulation and indicate in the methods the age of mice at surgery (8 weeks), behavioral testing (13-17 weeks), and perfusion and tissue collection. This study primarily served to identify cellular changes that do not necessarily require gene expression (such as DDR recruitment, ciliogenesis and PNN build up). Such changes were found with the Tlr9 KO and RelA KO even before CFC (data not shown, now indicated), so CFC-related changes could not be delineated from other factors, including spontaneous activity (Discussion, p.16, last pg). Because we did not expect such profound outcome and for comparison purposes, the KO genetic lines were matched for exposure and manipulations to the Tlr9KO that underwent CFC. We now specify this point (Results, p. 13, lines 7-10 from the bottom) and highlight that the inability to recruit DDR, or form cilia and PNN was a pre-existing state. Together with disruption of subsequent CFC-induced changes of gene expression, the data converge to a mechanism involving both CFC-dependent and independent regulation of centrosome function (genomic stability, DDR and ciliogenesis) by Tlr9. This issue needed clarification and we appreciate the comment.

4. Number of mice is not listed for experiments in Figure 4. Thus, it appears that the data shown is derived from one biological replicate.

This information is now provided. All data were obtained from multiple biological replicates (hippocampi of individual mice). We now indicate the numbers of neurons/mouse (when needed number of puncta per neuron), and number of mice/group for each experiment.

5. CFC behavior data in Figure 5 should include control experiments for both cued fear conditioning and short-term memory.

We now add information for both tone-fear conditioning and trace fear conditioning (impaired), which also replicated the context deficit found earlier. We replicated the Tlr9-induced deficit at least five times with different approaches (excluding the pharmacological manipulations). We did not add short-term memory data, however, because freezing is highly unreliable in mice in the short term after one-trial conditioning (which was preferred for detection of molecular changes). The cued experiment demonstrates that mice can acquire a freezing response despite the Tlr9 knockdown.

6. The neuronal specificity of the effect of Tlr9 deletion on CFC is supported using an astrocyte-specific Cre driver as a control (Figure 5b). Given the inflammatory

mechanisms downstream of Tlr9, the authors must include a microglia-specific Cre driver as an additional control.

Pease see general revisions.

7. Figure 1 (panels a, b, and d) is not legible in the resolution provided, which precludes full review of this manuscript.

New Figure1 is provided. We are sorry for the inconvenience.

Minor Concerns:

1. The analysis of Figure 3d appears to be at the level of individual neurons (this is unclear), although biological replicates for this experiment should be at the level of mice (n=6).

This is now clarified. We counted 60 neurons/mouse and then used the average to determine the group mean. This is why only 6 time points are shown in the graph. We opted for this analysis because presenting all neurons might have biased the outcome and show false significant results.

2. DNase2A shRNA in Figure 5d is driven by the U6 promoter, which would not result in a neuron-specific knockdown.

That is correct, this construct was selected because it proved most efficient in expressing DNase2 shRNA relative to other promoters and because we saw no evidence of dsDNA breaks in other cell populations. This was just a proof of principle experiment aiming to delineate Tlr9 from STING signaling more generally. We now present these data in the Extended Data Fig. 9.

3. AAV vectors for Syn-GFP and Syn-Cre are listed as serotype 9 in the text but serotype 8 in Figure 5a.

We apologize for the error, we used AAV9 in all experiments.

Reviewer 2

[REDACTED]

[REDACTED]

3. Fig 4s and 5. AAV9 syn-cre is used to remove TLR9 but is not used on control animals. This delivery method results in high expression of cre, which is known to cause extensive DNA damage characterized by gH2AX foci and breaks found in metaphase chromosome spreads (Silver and Livingston Mol Cell 2001 PMID11511376). This issue of directing cre recombinase in the floxed TLR9 mice but not in controls severely confounds the results. The appropriate control is AAV9 syn-cre in mice that do not contain floxed TLR9. This was not performed. For example, Fig 5h shows many more gH2AX foci in the syn-cre treated TLR9 floxed mice, which may simply reflect off target activity of syn-cre. Again, the more appropriate control would be to treat mice lacking a floxed allele with Syn-cre. These issues pose serious concerns about data interpretation.

Please see general revisions.

[REDACTED]

5. No independent methods are used to demonstrate that TLR9 actually directs DNA repair in neurons.

That is correct. All of our approaches are based on Tlr9 inhibition. Devising an opposite approach is not currently feasible in an *in vivo* system in which only a small number of cells undergo a physiological change. For clarification, we do not posit a direct role of Tlr9 in DDR but rather propose a role of a master coordinator of several

pathways including RelA and Dcx, that we discuss in detail, but also many other genes involved in nuclear export, and other relevant processes, as listed in the source data and supplementary tables. It is possible that the entire functional network rather than individual components are needed for Tlr9-initiated DDR, but we do not have a full answer at this time. We hope that Reviewer2 acknowledges the extent of future work needed to clarify all aspects of the demonstrated phenomenon.

6. A critical question is how TLR9 specifically recognizes cytosolic DNA and cGAS does not. Is there an endosomal context of dsDNA recognition that is responsible for this specificity? A simple staining experiment by immunofluorescence could be used to address this point.

It is well established that activation of Tlr9 by DNA fragments only takes place in the endomembrane system (vesicles), although recognition can take place in the ER, after which Tlr9 and DNA are further processed (through Hsp90b1, DNase2, etc) during vesicle trafficking. We noted that we frequently used the term “cytoplasmic”, which is misleading because it includes both vesicles and cytosol. We found no evidence of cytosolic release, and now provide further evidence for a perinuclear release of histone and DNA in Tlr9-rich, RNA-rich endomembrane compartments (Extended Data Fig. 3). We used the RNA label Syto rather than ER or vesicle labels for this study because perinuclear RNA synthesis predominantly takes place in the ER and because Syto also enabled us to identify nucleoli as potential site for dsDNA breaks (which was ruled out).

Reviewer 3

1. The authors perform bulk RNA sequencing of mouse hippocampi after Contextual Fear Conditioning and find differences in gene expression profiles after four days signifying an immune response. The data's bioinformatics analysis demonstrates the upregulation of TLR pathway genes and 'most prominently' CD14. It is well established that stress and CRC activate microglia cells, which trigger the migration of monocytes into the brain. Hence, it is likely that the observed upregulation of innate immune-related genes represents increased recruitment of monocytes to the brain rather than upregulation of TLR9 in neurons.

The authors need to demonstrate that CRC increases TLR9 and pathway components in neurons using a different approach, such as single-cell sequencing or spatial biologic methods.

Please see general revisions.

2. TLR9 is an endosomal protein that senses DNA in the endosomal compartment. Thus, there is a conceptual problem that needs to be addressed. How does TLR9 sense DNA damage in neurons in a cell-autonomous fashion, as proposed by the authors?

Our evidence points to perinuclear dsDNA and γ H2AX co-localization with Tlr9 following nuclear envelope rupture in ER/ribosome-rich compartments. But we cannot

rule out other mechanisms known to trigger Tlr9 responses to self-DNA, such as histones. This issue is now addressed in more depth (Discussion, p. 16, pg 1).

3. What is the evidence that TLR9 senses DNA damage in neurons? No direct data show that TLR9 activation occurs upon DNA damage. Only correlations are shown.

We are not aware of any *in vivo* study showing direct activation of Tlr9 by self-DNA fragments during a physiological process even in more accessible cellular systems such as the immune system. This is partly due to our inability to predict the identity of fragments that will be released during learning in an individual neuron and partly to our inability to manipulate their release. We show that activation of Tlr9 (enhanced *Tlr9* gene expression and localization in sites typical of DNA-induced activation) occurs in a neuronal context indicative of DNA damage (inflammatory gene expression, dsDNA breaks, extranuclear dsDNA, DDR). Extensive *in vitro* data have converged to DNA fragments as key activators of Tlr9. We have considered one alternative, that is histone, but that would also be consistent with our findings. We do not think that more direct evidence can be provided at this time, but instead rely on a body of previous *in vitro* data.

4. It is well established that AAVs are potent activators of type I production via mainly a TLR9-dependent mechanism. The in vivo work to reduce TLR9 expression in neurons is performed using AAV viral transduction of cells. Since AAVs can readily activate the TLR9-MyD88 pathway, this approach is highly prone to artifacts. The industry struggles with this issue for the translation of AAV-mediated gene delivery. Even the CpG content difference in the delivered genes versus controls might make a significant difference. Hence, this reviewer remains unconvinced by the neuron-specific deletion approach taken by the authors, as bystander cell TLR9 activation by the AAV vector could influence the phenotype.

See point 4, general revisions.

Minor issues:

1. The bioinformatics approach to data analysis remains obscure. Please describe the bioinformatic pipeline in detail. Fig. 1 a,b, and d are not legible and do not contain useful information. For example, for any TLR or other gene, similar hubs of genes could be generated, as shown in 1b. What is the relevance of showing this other than convincing the reader that this is the essential gene? On what basis has TLR9 been selected for highlighting?

This is now explained more extensively and the figures are revised to focus on the essential points (Fig. 1, Extended Fig. 1). We also improved dramatically the quality of the figure.

2. Imaging data require quantification.

All imaging data have been quantified and statistically analyzed.

Reviewer Reports on the First Revision:

Referees' comments:

Referee #1 (Remarks to the Author):

The authors have made revisions, both clarifications in the text and the inclusion of additional experiments, that strengthen the manuscript. The authors have convincingly clarified the conceptual novelty of their studies: they have uncovered a role for the pro-inflammatory Tlr9 signaling pathway in promoting memory formation by acting in neurons largely distinct from those undergoing IEG responses. We appreciate the efforts of the authors to clarify that Tlr9 signaling does not mark neurons committed to representing a single memory and that Tlr9 activation is not sufficient for memory. In addition, the technical novelty has also been improved by the inclusion of a single nuclei RNA sequencing experiment, which sheds light on the role of Tlr9 signaling in specific neuronal populations following CFC. The addition of a cued fear conditioning control experiment also strengthens the paper by demonstrating the contextual specificity of Tlr9 manipulations in disrupting memory.

However, there are two points arising from this revision that should be addressed.

1. The authors continue to interpret their data in a way that invokes the engram or suggests that reactivation of Tlr9 neurons is sufficient for memory. Examples of this interpretation include the title and the following passage from paragraph 4 of the discussion (lines 570-572):

Given the restricted reactivation of the population displaying an inflammatory phenotype during CFC and memory recall, this assembly most likely contributes to the stability of memory representations. This passage appears to refer to Figure 2f, which is insufficiently described in the text/figure legend and therefore difficult to interpret. Here the authors seem to claim to target neurons that exhibited Tlr9/inflammatory signaling during CFC and reactivate them during recall, which they do not. Our interpretation of the figure is that neurons activated/showing IEG responses during CFC training are more likely to show an IEG response, and less likely to exhibit DNA damage, during recall. This interpretation does support the idea that inflammatory responses/Tlr9 signaling occurs in neurons distinct from those undergoing IEG responses.

2. The inclusion of snRNA-seq adds crucial information regarding the cell-type specificity of the Tlr9 signaling response to CFC and additional molecular mechanisms contributing to this response, such as ER chaperones and regulators of vesicle function. However, the critical results from this experiment, such as those involving Hsp90b1, ATP6v0c, or Dcx, should be validated using a technique like RNAscope. The number of animals utilized for each group in this experiment is also not presented, casting doubt about the rigor of this experiment.

Referee #2 (Remarks to the Author):

The authors have addressed my primary concerns regarding cre induced damage artifact.
[REDACTED] Other issues have also been resolved.

Referee #3 (Remarks to the Author):

The reviewer acknowledges the authors' comprehensive efforts in addressing critiques. However, concerns persist regarding the activation of TLR9 by CFC in neurons and the mechanisms involved. Although the study observes an upregulation of TLR9 expression in bulk RNA seq upon CFC, the single-cell analysis provided by the authors does not definitively answer whether this upregulation occurs in neurons or results from myeloid cell influx into the brain. The absence of a clear response to this question raises the need for clarification. If technical limitations prevent the assessment of TLR9 mRNA levels, these challenges should be explicitly mentioned. Additionally, the authors could explore the possibility of analyzing myeloid cell ingress into the brain using single-cell datasets.

The authors argue that the colocalization of TLR9 with endosomal markers for late endosomes implies TLR9 activity in neurons. However, this claim is deemed incorrect, as TLR9 can autonomously traffic into various endosome subsets and only becomes activated in endosomes when ligands are present. Consequently, the evidence supporting TLR9 activation in neurons remains circumstantial.

A conceptual question regarding the cell-autonomous detection of DNA damage remains unaddressed. The reviewer seeks clarification on whether TLR9 in neurons directly recognizes DNA damage or if DNA damage results in the generation and release of TLR9-activating DNA fragments that neighboring neurons uptake. In the former case, the mechanism by which TLR9 in the endosome accesses damaged DNA, such as through autophagy, is not explained. In the latter case, where TLR9 triggers non-cell autonomous detection in bystander neurons, the rationale for this being a function of neurons rather than glial cell compartments is not clarified. Clarifying these points would enhance the overall understanding of the proposed concepts.

Author Rebuttals to First Revision:

Response to Reviewers

While remaining enthusiastic about our paper, the reviewers requested additional clarifications and validations. We have provided the requested information as detailed below. Because of the many added data, we had to remove all nonessential ED figures to comply with the Journal's requirements. Therefore, the Reviewers may note that some of the earlier figures are no longer presented. Nevertheless, all essential data, analyses, and quantifications are retained.

Reviewer 1

Point 1. *The authors continue to interpret their data in a way that invokes the engram or suggests that reactivation of Tlr9 neurons is sufficient for memory. Examples of this interpretation include the title and the following passage from paragraph 4 of the discussion (lines 570-572):*

Given the restricted reactivation of the population displaying an inflammatory phenotype during CFC and memory recall, this assembly most likely contributes to the stability of memory representations.

This passage appears to refer to Figure 2f, which is insufficiently described in the text/figure legend and therefore difficult to interpret. Here the authors seem to claim to target neurons that exhibited Tlr9/inflammatory signaling during CFC and reactivate them during recall, which they do not. Our interpretation of the figure is that neurons activated/showing IEG responses during CFC training are more likely to show an IEG response, and less likely to exhibit DNA damage, during recall. This interpretation does support the idea that inflammatory responses/Tlr9 signaling occurs in neurons distinct from those undergoing IEG responses.

Although we propose necessity, we do not suggest sufficiency with respect to the role of Tlr9 signaling in memory. Our framework is that different populations of CA1 neurons distinctly contribute to various aspects of memory, with populations displaying inflammatory signaling contributing to memory stability and persistence. The population displaying IEG responses, on the other hand, might be better suited for retrieval-mediated memory updates and modifications. This is now specified (Discussion, p. 16, pg2).

We also provide a better explanation for Fig 2f to make sure that we are on the same page as Reviewer 1. We completely agree that what we show is that IEG responsive neurons are more likely to up-regulate IEG (cFos) upon memory reactivation than neurons showing DDR. This supports findings obtained 1 h after CFC, showing low degree of overlap between neurons showing dsDNA breaks (nuclear γ H2AX signals) and IEG.

Point 2. *The inclusion of snRNA-seq adds crucial information regarding the cell-type specificity of the Tlr9 signaling response to CFC and additional molecular mechanisms contributing to this response, such as ER chaperones and regulators of vesicle function. However, the critical results from this experiment, such as those involving*

Hsp90b1, ATP6voc, or Dcx, should be validated using a technique like RNA scope. The number of animals utilized for each group in this experiment is also not presented, casting doubt about the rigor of this experiment.

We provide the requested validation with RNA scope for Hsp90b1 and Dcx (Extended Data Fig. 9) and add that the snRNA-seq data were obtained from pooled dorsal hippocampi of 5 mice/group (p. 10, last pg and Methods).

Reviewer 3

Point 1. *The reviewer acknowledges the authors' comprehensive efforts in addressing critiques. However, concerns persist regarding the activation of TLR9 by CFC in neurons and the mechanisms involved. Although the study observes an upregulation of TLR9 expression in bulk RNA seq upon CFC, the single-cell analysis provided by the authors does not definitively answer whether this upregulation occurs in neurons or results from myeloid cell influx into the brain. The absence of a clear response to this question raises the need for clarification. If technical limitations prevent the assessment of TLR9 mRNA levels, these challenges should be explicitly mentioned. Additionally, the authors could explore the possibility of analyzing myeloid cell ingress into the brain using single-cell datasets.*

We apologize for not sufficiently attending to several important issues raised by Reviewer 3. We now provide the results of the analysis of all key markers for immune cells in a new Extended Data Fig. 10, as a control for infiltrating cells as potential source of Tlr9 signaling. We did not find evidence for any lymphocytic/myeloid clusters. Expectedly, we found cells positive for macrophage/monocyte markers, however they were identified as microglia based on the presence of Siglech, Hexb, and Sall1. Thus, the strongest inflammatory phenotypes (revealed by conserved gene analysis, reactome analysis, and gene ontology analysis) were found in mature, Dcx⁻ CA and DG neurons.

Point 2. *The authors argue that the colocalization of TLR9 with endosomal markers for late endosomes implies TLR9 activity in neurons. However, this claim is deemed incorrect, as TLR9 can autonomously traffic into various endosome subsets and only becomes activated in endosomes when ligands are present. Consequently, the evidence supporting TLR9 activation in neurons remains circumstantial.*

We agree with Reviewer 3 that Tlr9 traffics autonomously. Our data support this view by showing the presence of Tlr9 in the ER and various endosomal populations. However, in various cell systems, Tlr9 localization in late endosomes was found to significantly increase after stimulation with CPG oligonucleotides leading to activation of Tlr9 (as first shown by Leifer et al., J Immunol 2004, <https://doi.org/10.4049/jimmunol.173.2.1179>; reviewed by Lee and Barton, Trends Cell Biol, 2014; Combes et al., Nat Commun, 2017). The similar increase induced by CFC, together with Tlr9-dependent RelA activation (which also depends on late endosomal Tlr9), strongly suggest enhanced activation of Tlr9.

We also agree and acknowledge in the paper that the evidence for Tlr9 activation based on up-regulation of *Tlr9* mRNA, extranuclear dsDNA release, and endosomal localization of Tlr9 is indirect. (p. 8, Discussion, 2nd pg). However, this is complemented by direct evidence for Tlr9-dependent coordination of neuron-specific inflammatory and DDR responses required for memory formation. We hope that Reviewer 3 agrees that such effects would not be possible without Tlr9 activation.

Point 3. *A conceptual question regarding the cell-autonomous detection of DNA damage remains unaddressed. The reviewer seeks clarification on whether TLR9 in neurons directly recognizes DNA damage or if DNA damage results in the generation and release of TLR9-activating DNA fragments that neighboring neurons uptake. In the former case, the mechanism by which TLR9 in the endosome accesses damaged DNA, such as through autophagy, is not explained. In the latter case, where TLR9 triggers non-cell autonomous detection in bystander neurons, the rationale for this being a function of neurons rather than glial cell compartments is not clarified. Clarifying these points would enhance the overall understanding of the proposed concepts.*

To the best of our knowledge, there is no evidence for the involvement of Tlr9 in the detection of DNA damage (because Tlr9 is mostly compartmentalized outside of the nucleus and we never saw co-localization of nuclear γ H2AX and Tlr9 signals) but only for detection of extranuclear DNA fragments. However, these fragments are typically recognized cell autonomously (as they reach the endosomal compartments), without being released extracellularly (e.g., Barber GN. *Cytoplasmic DNA innate immune pathways. Immunol Rev. 2011*). Nevertheless, we agree with Reviewer 3 that addressing this issue is important, and therefore performed additional experiments using treatment with DNase1 (Extended Data Fig. 10). This enzyme efficiently degrades cell free, circulating and extracellular DNA, and if such DNA fragments contributed to the memory effects that we observed, we would have found evidence for memory impairment. This, however, was not the case.

With this in mind and given that we did not detect any dsDNA breaks in microglia or astrocytes, and that most breaks in response to CFC were induced in neurons, we posit a model of memory-related neuronal activation that induces dsDNA damage, release of dsDNA fragments in the endomembrane subcellular compartments, and cell autonomous activation of Tlr9. This is followed by NF- κ B activation, centrosomal DDR, ciliogenesis, and PNN formation. While the activation steps need further interrogation, we do provide direct evidence for the consequences of Tlr9 activation (including NF- κ B activation, centrosomal DDR, ciliogenesis, and PNN formation) relevant for memory.

Lastly, we would like to acknowledge that our work does not exclude the roles of Tlr9 signaling in microglia, astrocytes, immune, and other non-neuronal cell types in brain function. On the contrary, we fully agree with Reviewer 3 that extracellular release of DNA fragments and non-neuronal Tlr9 signaling could play prominent roles in pathophysiological processes including apoptosis, neurodegeneration, and infective

brain diseases. However, our work's main focus was on a physiological process, memory formation, which did not result in any abnormalities that would lead to non-cell autonomous microglial, astrocytic, or blood born immune activation. We therefore retained focus on neuron-specific Tlr9 signaling throughout the manuscript and hope that Reviewer 3 will find our approach reasonable.

Reviewer Reports on the Second Revision:

Referees' comments:

Referee #1 (Remarks to the Author):

The authors have appropriately addressed our concerns described in previous reviews.

There is one important question about the snRNA seq experiment. In the current version of the manuscript and in the rebuttal, the authors state "the snRNA-seq data were obtained from pooled dorsal hippocampi of 5 mice/group (p. 10, last pg and Methods)." Does this then mean that the snRNA-seq experiment effectively has an n of 1 per group? If so then additional samples need to be sequenced to better account for variability.

Referee #3 (Remarks to the Author):

This authors' responses have addressed each of the reviewer's requests. I am now satisfied with the revisions made to the manuscript and support publication of the manuscript.

Author Rebuttals to Second Revision:

Reviewer 1

There is one important question about the snRNA seq experiment. In the current version of the manuscript and in the rebuttal, the authors state "the snRNA-seq data were obtained from pooled dorsal hippocampi of 5 mice/group (p. 10, last pg and Methods)." Does this then mean that the snRNA-seq experiment effectively has an n of 1 per group? If so then additional samples need to be sequenced to better account for variability.

Reviewer 1 has expressed a concern regarding the n/group for the snRNAseq data. We acknowledge that an n of 1 is used for sequencing for every condition, however, each condition contains ~8000 nuclei from 5 biologically different samples (~1600 nuclei/mouse). We faced a choice between having sufficient number of cells vs sufficient number of biological replicates and the former prevailed given that we did not know which cell cluster would be most affected. This choice was also based on prior support for the validity of sample pooling for snRNAseq (Assefa et al., 2020; Kalucka et al., 2020). The consistency of cell clusters across groups and homogeneity of gene responses within single clusters (ED Fig. 10) strongly suggests that the observed effects were present in nuclei from all samples. Lastly, we provided validation with RNAscope using additional 4 biological samples for the main genes of interest.

Takele Assefa, A., Vandesompele, J. & Thas, O. On the utility of RNA sample pooling to optimize cost and statistical power in RNA sequencing experiments. *BMC Genomics* **21**, 312 (2020).

Kalucka J, de Rooij LPMH, Goveia J, Rohlenova K, Dumas SJ, Meta E, Conchinha NV, Taverna F, Teuwen LA, Veys K, García-Caballero M, Khan S, Geldhof V, Sokol L, Chen R, Treps L, Borri M, de Zeeuw P, Dubois C, Karakach TK, Falkenberg KD, Parys M, Yin X, Vinckier S, Du Y, Fenton RA, Schoonjans L, Dewerchin M, Eelen G, Thienpont B, Lin L, Bolund L, Li X, Luo Y, Carmeliet P. Single-Cell Transcriptome Atlas of Murine Endothelial Cells. *Cell*. 2020 Feb 20;180(4):764-779.e20. doi: 10.1016/j.cell.2020.01.015. Epub 2020 Feb 13. PMID: 32059779.